# Diversity within the adenovirus fiber knob hypervariable loops influences primary receptor interactions

Alexander T. Baker[1], Alexander Greenshields-Watson[2], Lynda Coughlan [3], James A. Davies[1], Hanni Uusi-Kerttula[1], David K. Cole[2,4], Pierre J. Rizkallah[2] & Alan L. Parker [1]

Adenovirus based vectors are of increasing importance for wide ranging therapeutic applications. As vaccines, vectors derived from human adenovirus species D serotypes 26 and 48 (HAdV-D26/48) are demonstrating promising efficacy as protective platforms against infectious diseases. Significant clinical progress has been made, yet definitive studies underpinning mechanisms of entry, infection, and receptor usage are currently lacking. Here, we perform structural and biological analysis of the receptor binding fiber-knob protein of HAdV-D26/48, reporting crystal structures, and modelling putative interactions with two previously suggested attachment receptors, CD46 and Coxsackie and Adenovirus Receptor (CAR). We provide evidence of a low affinity interaction with CAR, with modelling suggesting affinity is attenuated through extended, semi-flexible loop structures, providing steric hindrance. Conversely, in silico and in vitro experiments are unable to provide evidence of interaction between HAdV-D26/48 fiber-knob with CD46, or with Desmoglein 2. Our findings provide insight into the cell-virus interactions of HAdV-D26/48, with important implications for the design and engineering of optimised Ad-based therapeutics.

[1] Division of Cancer and Genetics, School of Medicine, Cardiff University, Cardiff CF14 4XN, UK. [2] Division of Infection and Immunity, School of Medicine, Cardiff University, Cardiff CF14 4XN, UK. [3] Department of Microbiology, Icahn School of Medicine at Mount Sinai, New York, NY 10029-6574, USA. [4]Present address: Immunocore Ltd., Abingdon OX14 4RY Oxon, UK. Correspondence and requests for materials should be addressed to A.L.P. (email: ParkerAL@Cardiff.ac.uk)

Adenoviruses are increasingly important vectors for wide-ranging therapeutic interventions, from gene delivery and oncolytic agents to platforms for vaccine applications[1–3]. As vaccine vectors, their use clinically has been popularised by their excellent safety profile coupled with their ability to induce robust cellular and humoral immunogenicity in humans[4]. Phylogenetically, the human adenoviruses (HAdV's) are diverse, subdivided across seven species, A–G[5], based classically on serological cross-reactivity, receptor usage, haemagglutination properties and, more recently, phylogenetic sequence similarity[6,7].

Most experimental and clinical studies have focussed on the well-studied species C adenovirus, HAdV-C5. Although potently immunogenic, the efficacy of vaccines based on HAdV-C5 appears hampered by high seroprevalence rates in humans, and enthusiasm for their use as clinical vaccine platforms has been dampened by the well-publicised failure of the MERCK sponsored STEP vaccine trial. This trial, to evaluate an HAdV-C5-based HIV vaccine encoding HIV gag/pol/nef antigens, was abandoned due to apparent lack of efficacy upon 1st term analysis. The study also identified a non-significant trend towards increased HIV acquisition in a specific high-risk, uncircumcised subset of patients who also had high levels of baseline pre-existing neutralising antibodies (NAbs) to HAdV-C5[8,9]. As a result, attention has switched from HAdV-C5-based vectors towards the development of alternative adenoviral serotypes with lower rates of pre-existing immunity. Most notably, vectors under development include those based on species D serotypes including HAdV-D26, which has entered Phase-III clinical trials as an Ebola vaccine and recently reported promising immunogenicity in an HIV trial. Chimeric vectors utilising the hexon hyper variable regions (HVRs) of HAdV-D48 which have undergone Phase-I evaluation as an HIV vaccine[3,10–12]. However, despite extensive clinical advances using these vaccine vectors we possess very limited knowledge of their basic biology, particularly with regards to the determinants underpinning their tropism, mechanisms of cellular entry, and receptor usage. In this study, we address these shortcomings through analysis of adenoviral diversity in the context of their receptor binding, fiber proteins. Whilst adenoviruses are historically divided into seven species, A–G, this may underestimate their diversity[13–15]. Phylogenetic examination of 56 human adenovirus fiber proteins from different species shows deviation from the taxonomy expected based upon whole genome taxonomy, likely due to recombination events as seen in other adenoviral proteins[14]. Here, we have sought to generate high-resolution crystallographic structures of the cellular interacting fiber-knob domains of species D adenoviruses HAdV-D26 and HAdV-D48. The fiber-knob is the receptor interacting domain of the fiber protein, one of three major capsid proteins along with the hexon and penton base, as shown schematically in Fig. 1a.

In this study, we employ an integrative workflow utilising X-ray crystallography, in silico modelling, and in vitro assays to dissect previous findings[16,17] suggesting interactions by HAdV-D26 and HAdV-D48 with Coxsackie and Adenovirus Receptor (CAR)[5,18] and CD46 (Membrane Cofactor Protein, MCP)[16,17,19–21]. Utilising surface plasmon resonance (SPR), we also investigate the potential for HAdV-D26 and HAdV-D48 to interact with Desmoglein 2. Our findings shed new light on the cell–virus interactions of adenovirus and have potential implications for the design and engineering of optimised HAdV-based therapeutics, both for vaccine applications and oncolytic development, allowing us to minimise off-target or undesirable interactions in vivo.

## Results

**Genetic variability in adenovirus fiber-knob protein.** We generated phylogenetic trees of human adenovirus serotypes 1–56 which revealed greater diversity in the fiber-knob domain (Fig. 1b) than might be expected based upon the taxonomy of the whole virus (Fig. 1c). These phylogenetic trees have been condensed to 70% bootstrap confidence (500 bootstrap replicates) to exclude poorly supported nodes and display the projected diversity. A full dendrogram showing to-scale branches is provided in Supplementary Figure 1. In both phylogenetic trees, the adenoviruses divide into seven clades corresponding to the seven adenoviral species, A–G. However, the tree based upon the fiber-knob domain (Fig. 1b) shows the species D adenoviruses forming a greater number of sub-groups than in the whole genome tree suggesting greater diversity in the receptors of this species than might be expected when comparing serotypes. This may be the result of recombination events, as reported previously for other adenoviral proteins[13,15,22–24]. The opposite is observed in species B adenoviruses, where simpler groupings are seen when analysing fiber-knob domain alignment than by whole genome analysis. When divided into sub-species based on whole genome, they divide into species B1 and B2, but when the tree is generated based on fiber-knob alone the species B viruses do not divide into similar groups. The significance of this fiber-knob diversity is unclear, but it has previously been suggested that the species B1/B2 designation may more closely represent the tissue tropism, than receptor usage[25,26].

We next calculated the amino acid variability at each position in the aligned adenoviral fiber-knob sequences, which revealed regions of broad conservation corresponding to β-strands which make up the main fold of the fiber-knob trimer (Fig. 1d). The positions corresponding to the β-strands of HAdV-C5, as originally reported by Xia et al.[27], are shown by arrows. Comparison shows that the more N-terminal (A, B, and C) and C-terminal (I and J) β-strands have greater homology across the adenoviral species than the other strands. This may relate to the intervening loops between the less tightly conserved β-strands (D, E, F, G, H) being more apical, a region which is often involved in receptor interactions[26,28–32].

**Structural analysis of HAdV-D26 and HAdV-D48 fiber-knob.** To investigate diversity within the species D adenoviruses fiber-knob protein, recombinant, 6-His-tagged fiber-knob protein from HAdV-D26 and HAdV-D48 (hereafter referred to as HAdV-D26K and HAdV-D48K) were generated, purified, and used to determine X-ray crystallographic structures of HAdV-D26 (PDB: 6FJN) and HAdV-D48 (PDB: 6FJQ) at resolutions of 0.97 and 2.91 Å, respectively (Fig. 2a–d). Table 1 shows the data collection and refinement statistics for the crystallographic structures generated in this study.

The monomers (red, green, blue) form an anti-parallel β-barrel, typical of adenoviral fiber-knob protein, as described by Xia et al. (PDB 1KNB)[27]. Each monomer interacts with two neighbouring copies to form a homotrimer with 3-fold symmetry (Fig. 2b, d) and a highly stable interface (Supplementary Figure 2A). Stability analysis using PISA (Protein Interactions, Surfaces and Assemblies) software calculates the HAdV-D26 and HAdV-D48 trimers to have >20% lower interface energy than that of HAdV-C5, indicative of a more stable interaction (Supplementary Figure 2A) between the monomers of the HAdV-D26 and HAdV-D48 fiber-knobs.

As with the pan species analysis (Fig. 1d), variability of the aligned species D fiber-knobs (Fig. 2e) confirmed that the β-strands comprising the hydrophobic core of the trimers are

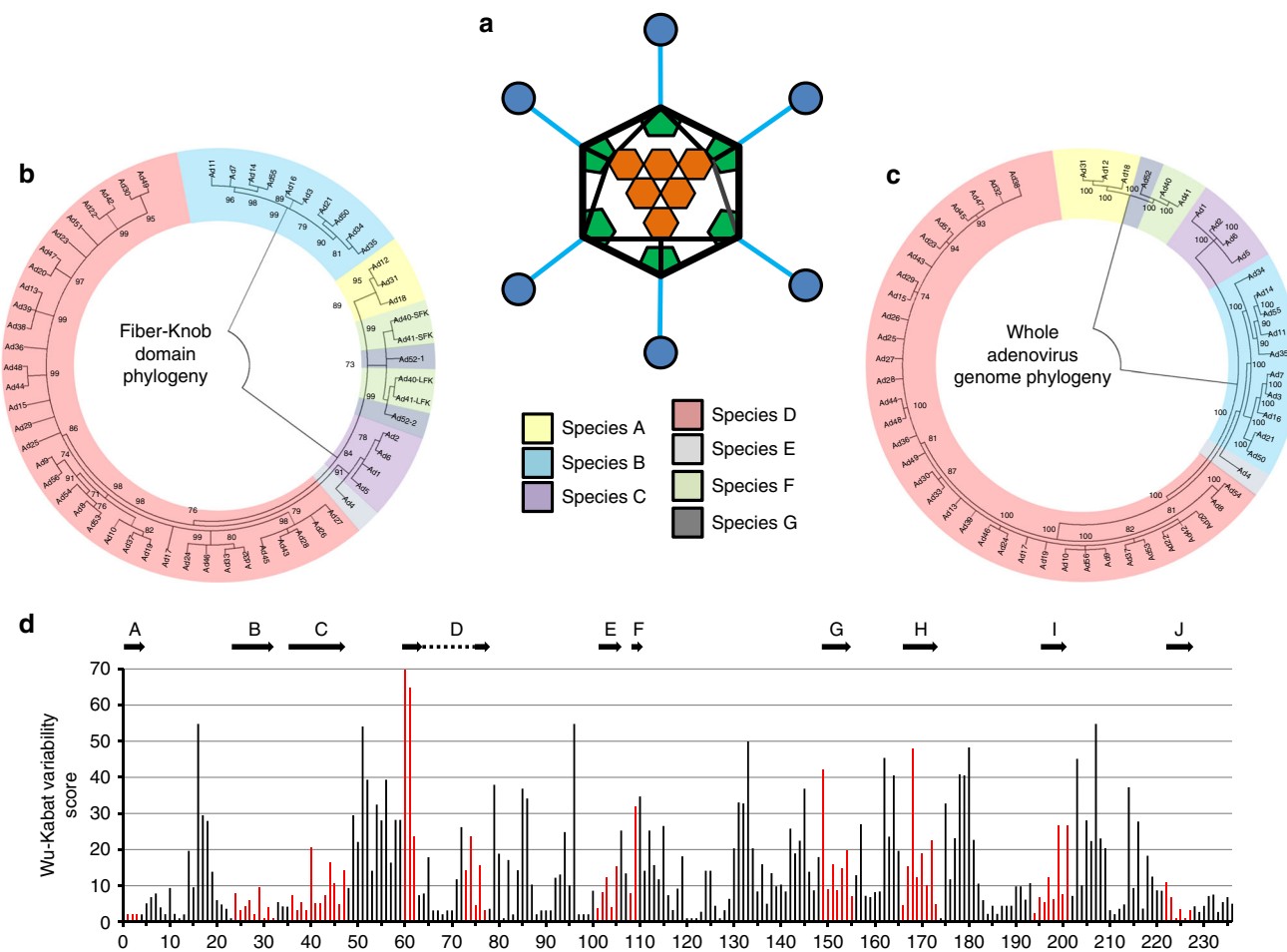

**Fig. 1** Phylogenetic analysis of adenoviruses mapped by whole genome and fiber-knob domain. A diagrammatic representation of the adenoviral major capsid proteins shows the icosahedral capsid structure with the Hexon (orange) comprising the facets, pentons (green) at the vertices, from which the fiber proteins (fiber-shaft in light blue, fiber-knob in dark blue) protrude (**a**). Condensed maximum likelihood trees (percentage confidence shown by numbers next to nodes) were generated from alignments of fiber-knob domain amino acid sequences of adenoviruses 1–56 (**b**) or whole genome NT sequences (**c**). Adenoviruses divide into 7 subspecies, as denoted in the key, regardless of alignment used, but the species D adenoviruses divide into additional sub-species when determined by fiber-knob alignment, for readability simple nomenclature is used, all are human adenovirus. Numbers next to nodes denote confidence. Wu–Kabat variability analysis of the Clustal Omega aligned fiber-knob domains amino acid sequences of adenoviruses 1–56 (**d**) reveals regions of low amino acid variability corresponding to beta-sheets. The locations of HAdV-C5 β-strands, as described by Xia et al. (1994)[27], are aligned to the structure and shown by arrows, the corresponding positions are coloured in red

highly conserved, with β-strands demonstrating a high degree of overlap in both spatial position and sequence variability (Supplementary Figure 2B).

**Fiber-knob loops are stabilised by inter-loop interactions.** Particularly relevant to this study are the DG, GH, HI, and IJ loops, linking the indicated strands corresponding to those in the originally reported HAdV-C5 fiber-knob loops (Supplementary Figure 2B)[27]. These loops have previously been shown to be critical in engagement of CD46 for Ad11, 35, and 21[33–36]. Alignment of these loops with the corresponding loops of HAdV-C5 (species C, CAR interacting), HAdV-B35 (species B, CD46 interacting), Ad11 (species B, CD46 interacting), and Ad37 (species D, CAR and Sialic acid/GD1a glycan interacting) reveals different topologies in these critical receptor interacting regions (Fig. 3a)[35–38]. The HI loops of HAdV-D26K and HAdV-D48K are most homologous to those of HAdV-B35K and HAdV-D37K respectively in terms of amino acid sequence identity (Supplementary Figure 2B) and spatial alignment. The HAdV-D26K DG loop is most homologous to HAdV-B35K but incorporates a

three amino acid insertion, while HAdV-D48K DG loop displays a differing and unique topology. The GH and IJ loops of HAdV-C5K, HAdV-D26K, HAdV-D37K, and HAdV-D48K demonstrate similar spatial arrangements (though the similarity does not extend to the sequence identity) but differ from the CD46 utilising HAd-B11K and HAdV-B35K.

The high-resolution structures obtained allowed us to robustly characterise the loops, seen in the electron density maps (Fig. 3b, c). To assess loop flexibility and mobility, we assessed the B-factors (also known as temperature factor), a measure of the confidence in the position of an atom which can be used to infer flexibility. By assessing the B-factors, the relative flexibility of the moieties of interest can be inferred. While the apical domains of some loops demonstrated increased B-factors relative to the rest of the molecule, the loops' B-factors are surprisingly low (Fig. 3b, c), suggesting that they may exhibit limited flexibility.

To investigate whether the different loop conformations were the product of flexibility, or restricted mobility we investigated the inter-loop interactions in the HAdV-D26K and HAdV-D48K structures. This analysis shows that the GH loop of HAdV-D26K (like those of HAdV-C5K, HAdV-D37K, and HAdV-D48K) does

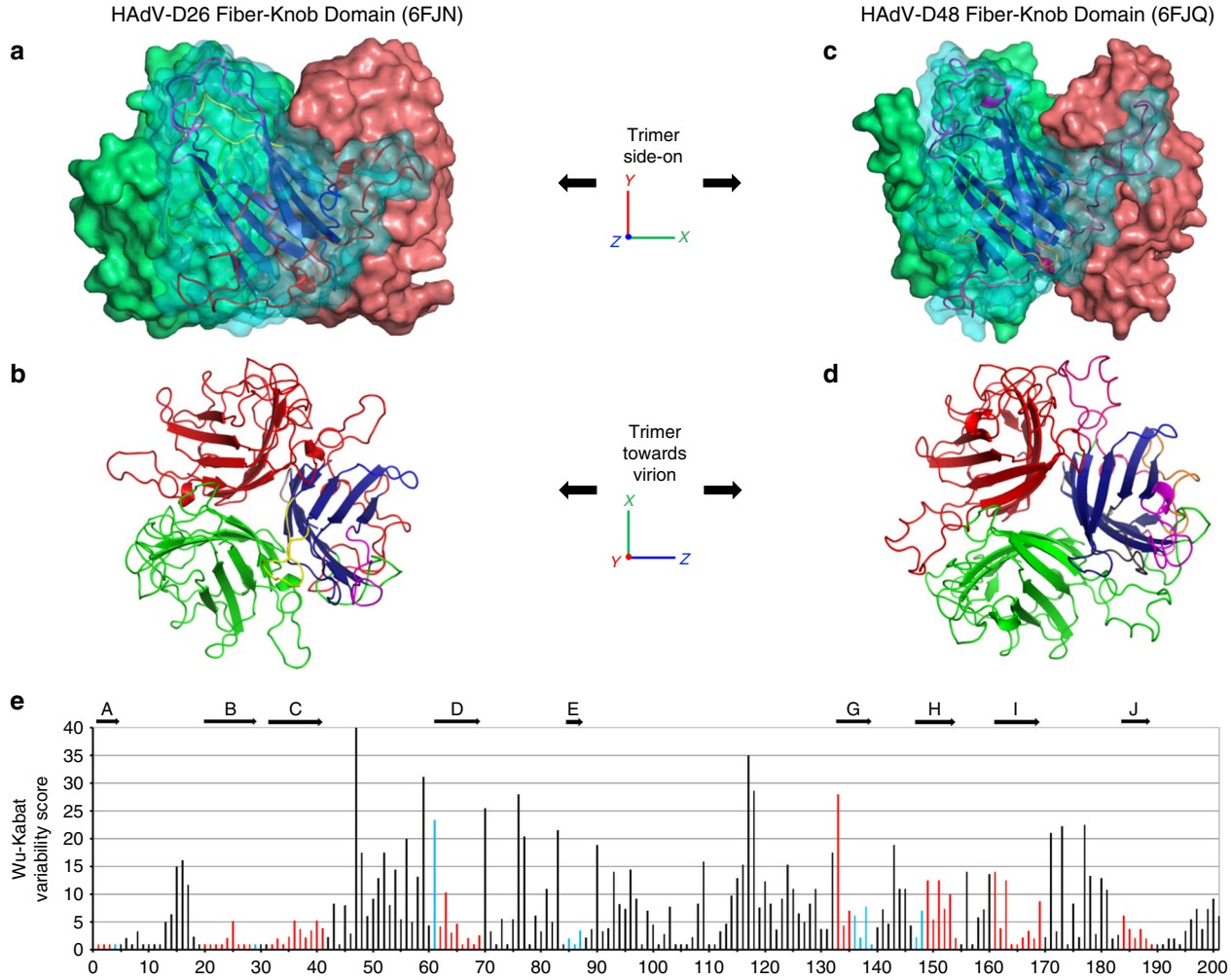

**Fig. 2** Overview of the HAdV-D26 and HAdV-D48 fiber-knob protein structures. The surface representation of the trimeric HAdV-D26K (PDB 6FJN) biological assembly is shown side-on with the cartoon representation shown for the nearest monomer (**a**) and the top-down view of the same HAdV-D26K trimer, as it would appear looking towards the virion, is seen as a cartoon representation (**b**), with each monomer coloured in red, green, or blue with the hypervariable loops extending between the β-strands (dark blue) coloured as follows: AB—green, BC—purple, CD—brown, DE—orange, DG—red, EG—pink, GH—purple, HI—yellow, IJ—light blue. The HAdV-D48K (PDB 6FJQ) trimer is shown similarly (**c**, **d**). The Wu–Kabat variability plot of the fiber-knob domains of species D adenoviruses shows regions of low variability (**e**) with the locations of the HAdV-D48K β-strands shown by arrows above the graph, and the positions coloured blue, or red when the position is a β-strand in both HAdV-D26K and HAdV-D48K

not extend directly away from the G and H β-sheets, but forms a β-hairpin (Fig. 3a, b) that is maintained by seven polar contacts within the neighbouring IJ loop which restrict the loops' orientation (Fig. 4a, Supplementary Figure 3A). Polar contacts were also observed at the apex of several loops, notably the GH and CD loops (Fig. 4a, b). The IJ loops form fewer intramolecular polar contacts but are stabilised by interactions with the adjacent CD and DG loops (Supplementary Figure 3C, D). These interactions retain the apical residues in a stable conformation, rather than leaving the side chains fully labile.

The B-factors of the HAdV-D48K DG loop were observed to be polarised about the hairpin, with the outer face of the loop having higher B-factors compared to the inner face (Fig. 4c). This is likely the result of polar contacts formed between Ser-307, Gln-308, Ala-309, and Leu-304 with Asp-359 and Gln-357 of the opposing monomer stabilising the conformation of the DG loop. The proline-rich nature of this loop provides further rigidity (Supplementary Figure 2D).

Crystal contacts did not reveal any specific interactions between these DG-loops and neighbouring non-trimer copies. We calculated the energy of interaction to be below the

background threshold ($<-3.0\,\text{kcal mol}^{-1}$) for all loops except DG. The DG-loop of HAdV-D26K is calculated to have interaction energy of $-6.5\,\text{kcal mol}^{-1}$ in two separate stretches of this exceptionally long loop (Supplementary Figure 4). Importantly, no strong contacts are found within the inter-monomer cleft.

Based on this analysis of the inter and intra-loop bonds, we suggest that these adenoviral loops may not be fully flexible variable regions, but organised receptor engagement motifs with carefully evolved structures. This has direct implications for receptor engagement of these viruses, as the loops govern previously characterised interactions with CAR and CD46, and are directly involved in their pathogenicity[25,33,39].

**In silico evaluation of HAdV-D26/48K interaction with CAR.** Both CD46 and CAR have been proposed as primary attachment receptors for HAdV-D26 and HAdV-D48[16,17]. Previously generated crystal structures of HAdV-B11K in complex with full-length CD46 (PDB: 3O8E), and HAd-D37K in complex with CAR-D1 domain (PDB: 2J12) reveal the loops to be essential to

**Table 1 Data collection and refinement statistics (molecular replacement)**

|  | HAdV-D26K | HAdV-C5K | HAdV-D48K |
|---|---|---|---|
| **Data collection** | | | |
| Space group | P 2$_1$3 | P 2$_1$2$_1$2 | P 4$_3$32 |
| Cell dimensions | | | |
| $a$, $b$, $c$ (Å) | 86.01, 86.01, 86.01 | 102.16, 102.44, 77.01 | 145.18, 145.18, 145.18 |
| $\alpha$, $\beta$, $\gamma$ (°) | 90.0, 90.0, 90.0 | 90.0, 90.0, 90.0 | 90.0, 90.0, 90.0 |
| Resolution (Å) | 0.97–60.82 (0.97–1.00) | 1.49–61.56 (1.49–1.53) | 2.91–83.82 (2.91–2.99) |
| $R_{sym}$ or $R_{merge}$ | 0.043 (0.745) | 0.134 (1.838) | 0.125 (302.6) |
| $I/\sigma I$ | 27.3 (0.7) | 7.1 (0.7) | 22.2 (1.7) |
| Completeness (%) | 94.9 (43.9) | 99.8 (99.9) | 100.0 (100) |
| Redundancy | 16.7 (1.6) | 6.6 (6.3) | 41.2 (41.4) |
| **Refinement** | | | |
| Resolution (Å) | 0.97–60.82 | 1.49–61.56 | 2.91–83.82 |
| No. reflections | 112,612 | 125,479 | 11,371 |
| $R_{work}/R_{free}$ | 18.2/19.5 | 21.1/23.3 | 20.1/29.1 |
| No. atoms | 1811 | 4825 | 3117 |
| Protein | 1579 | 4395 | 3091 |
| Ligand/ion | 8 | 21 | 20 |
| Water | 224 | 409 | 6 |
| $B$-factors | 16.0 | 34.0 | 87.0 |
| Protein | 15.8 | 33.6 | 94.2 |
| Ligand/ion | 29.3 | 35.7 | 129.5 |
| Water | 23.9 | 42.9 | 61.0 |
| R.M.S. deviations | | | |
| Bond lengths (Å) | 0.025 | 0.011 | 0.019 |
| Bond angles (°) | 2.339 | 1.534 | 2.293 |

One crystal was used for each dataset
Values in parentheses are for highest-resolution shell

receptor interactions[40]. To investigate the ability of HAdV-D26K and HAdV-D48K to bind these receptors we generated homology models by alignment of the new HAdV-D26K and HAdV-D48K fiber-knob structures modelled, using the existing fiber-knobs in complex with the receptor of interest, and performed energy minimisation to optimise the conformation to achieve the lowest possible energy interface with which to analyse the interaction. We performed similar experiments with the well-described CAR and CD46 binding fiber-knob proteins of HAdV-C5 (PDB: 6HCN) and HAdV-B35 (PDB: 2QLK), respectively, as controls.

Modelling of HAdV-D26K and HAdV-D48K in complex with the CAR-D1 domain revealed a region of high homology with the CAR utilising HAdV-C5 fiber-knob, hereafter termed the α-interface (Fig. 5a). Sequence alignment with HAdV-C5K shows that many of the residues previously shown to be critical for CAR interaction in HAdV-C5K are conserved in HAdV-D26K and HAdV-D48K (Fig. 5b), including Ser-408, Pro-409, and Tyr-376[37]. The same is true of residues predicted to interact with CAR directly, such as Lys-417 (number is for HAdV-C5K). The residues predicted to form direct CAR binding interactions for HAdV-C5K, HAdV-D26K, and HAdV-D48K are pictured in complex with the maximum spatial occupancy of the energy minimised CAR-D1 (Fig. 5c). The high levels of homology are seen to extend to the proteins' fold as well as the linear sequence.

Binding energies were calculated between the modelled fiber-knob proteins and CAR, restricting the calculation to only the α-interface to best model the conserved region. For the modelled complexes, a stable α-interface was predicted for all complexes modelled, albeit weaker for the known non-CAR utilising HAdV-B35 (Fig. 5d) which has lower sequence conservation with HAdV-C5K. However, the interaction is complicated by a second CAR interface, termed the β-interface (Fig. 6a). The loops

forming the β-interface with CAR-D1 differ between HAdV-C5, HAdV-D26, and HAdV-D48 fiber-knob (Fig. 6b). The shorter HAdV-C5K DG loop does not clash with the CAR-D1 surface, whereas the extended HAdV-D26K DG loop forms a partial steric clash, with surface seen to clash with the aligned CAR-D1, and HAdV-D48K DG loop is seen to form an even larger steric clash. Whilst the longer loop of HAdV-D26K is expected to be more flexible than that of HAdV-C5K, the HAdV-D48K DG loop is surprisingly stable due to the characteristics described (Fig. 4c).

**Biological evaluation of HAdV-D26/48K interaction with CAR.** Our modelling studies indicate that the longer, inflexible DG loop of HAdV-D48K would be likely to sterically hinder the HAdV-D48K: CAR interaction at the β-interface to a greater extent than the more modest inhibition of the smaller and more labile loop of HAdV-D26K, which in turn would exhibit more inhibition of CAR binding than that of HAdV-C5K, where no steric inhibition is observed. Competition inhibition assays using recombinant fiber-knob protein to inhibit antibody binding to CAR receptor in CHO-CAR cells (which express CAR, while the parental cell line (CHO-K1) is established to be non-permissive to adenovirus infection) support our observations (Fig. 6c). The IC$_{50}$ (the concentration of protein required to inhibit 50% of antibody binding) of HAdV-C5K is 7.0 ng/10$^5$ cells, while HAdV-D26K and HAdV-D48K demonstrate IC$_{50}$ values 15.7 and 480 times higher at 0.110 μg/10$^5$ cells and 3.359 μg/10$^5$ cells, respectively, reflecting their reduced ability to engage CAR.

SPR analysis indicates that HAdV-C5K binds strongly to CAR (Fig. 6d) with a $K_D$ of 0.76 nM. HAdV-D26K and HAdV-D48K have lower overall affinities for CAR (Fig. 6e). While the $K_{Off}$ of the three fiber knob proteins (Fig. 6e) are similar, the $K_{On}$ is fastest for HAdV-C5K, with HAdV-D26K $K_{On}$ being slower, and HAdV-D48K exhibiting the slowest $K_{On}$. This shows that the $K_{On}$—the ability to form the initial interaction with the receptor—is the major limiting factor in the fiber knobs overall affinity for CAR.

**In silico evaluation of HAdV-D26/48K interaction with CD46.** A similar approach was taken to model HAdV-D26K and HAdV-D48K in complex with CD46. Alignments with the previously published HAdV-B11K-CD46 complex were generated and energy minimised to obtain the lowest energy state of the complex[34,41]. This interface utilises loops DG, GH, HI, and IJ to form a network of polar interactions with the CD46 Sc1 and Sc2 domains (Fig. 7a)[35].

When HAdV-B35K was modelled in complex with CD46, a network of polar contacts between HAdV-B35K and CD46 was predicted (Fig. 7b) similar to that observed in the HAdV-B11K-CD46 complex crystal structure, PDB: 3O8E (Fig. 7a)[34]. Previously, HAdV-B35K residues Phe-243, Arg-244, Tyr-260, Arg-279, Ser-282, and Glu-302 (underlined in Fig. 7c) have been implicated as key contact residues for CD46 interaction, and are conserved in HAdV-B11K (highlighted in blue, Fig. 7c)[36]. Our modelling suggests that conversely, HAdV-D26K and HAdV-D48K are predicted to form very few polar contacts with CD46 with just two contacts predicted for HAdV-D26K (Fig. 7d) and three predicted for HAdV-D48K (Fig. 7e). Furthermore, they do not share any of the critical CD46 binding residues which have been reported previously (underlined, Fig. 7c) for HAdV-B11K and HAdV-B35K, or any of the predicted interacting residues (blue highlight, Fig. 7c).

We again employed PISA to calculate the binding energy of the various modelled and energy minimised fiber-knob CD46 complexes (Fig. 8a). HAdV-B11K, the strongest known CD46 binding adenovirus[33], was predicted to have the lowest binding

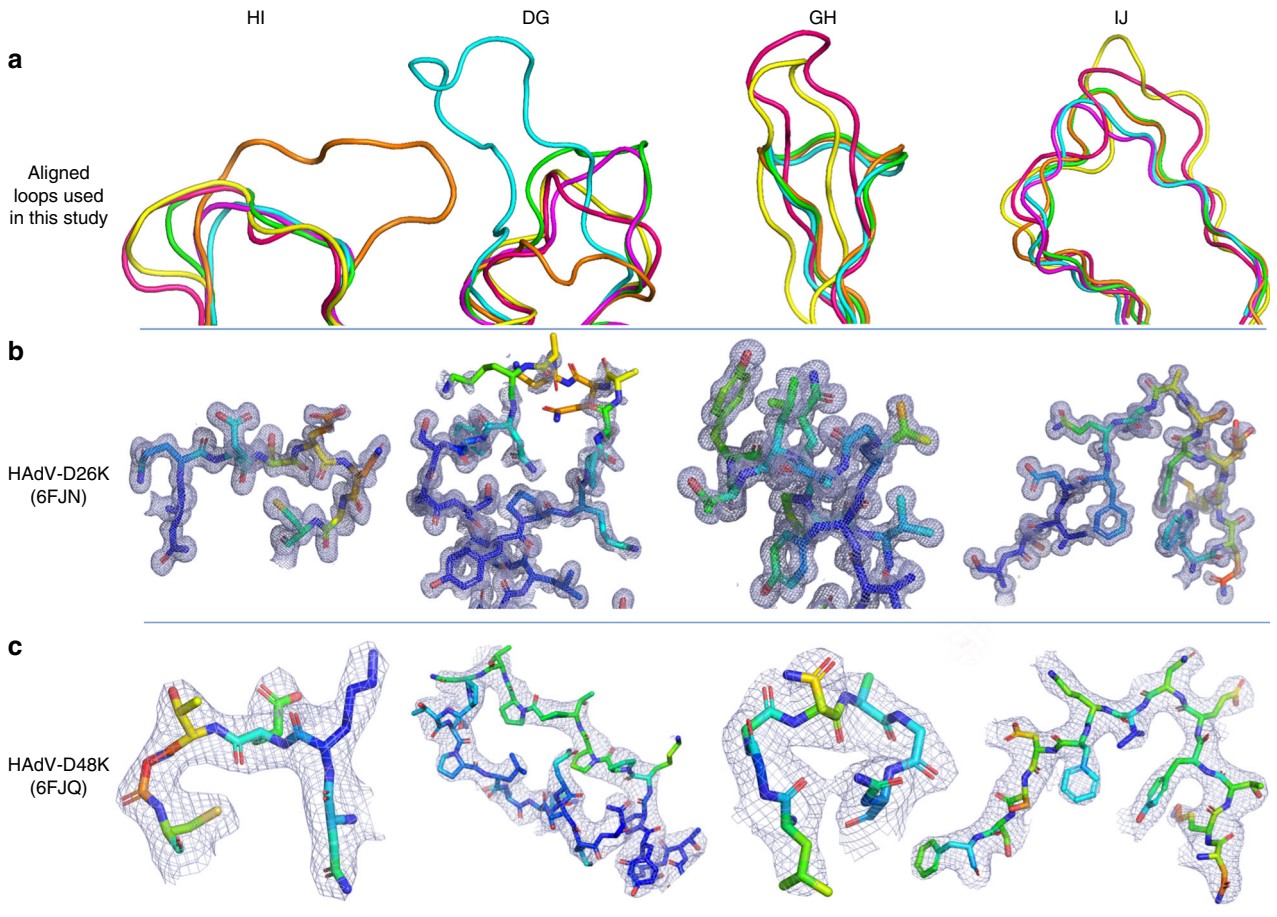

**Fig. 3** Comparison of HI, DG, GH, and IJ loops of adenoviruses used in this study. **a** The hypervariable loops of HAdV-D26K (green) and HAdV-D48K (cyan) relevant to this study (HI, DG, GH, and IJ) are shown in the context of the control virus fiber-knob domains, HAdV-C5K (orange), Ad11K (yellow), HAdV-B35K (pink), and Ad37K (purple). The electron density achieved in the loops of HAdV-D26K (**b**) and HAdV-D48K (**c**) are shown as mesh. The fitted residues are seen as stick representations, with oxygen and nitrogen atoms coloured red and blue, respectively and other atoms coloured according to their relative B-factors with warmer colours indicating higher B-factor values

energy reflecting its high stability interface, with HAdV-B35 demonstrating a similar but slightly reduced binding energy. Conversely, HAdV-D26K and HAdV-D48K are predicted to have lower binding energies, similar to that which may be expected for random proteins passing in solution, indicating that any interaction between either the HAdV-D26K, or HAdV-D48K with CD46 is unlikely[42].

While still low compared to the known CD46 interacting HAdV-B11K and HAdV-B35K binding energies, that for HAdV-C5K was higher than expected for a known non-CD46 interacting adenovirus (Fig. 8a). Inspection of the model shows that this is due to the close proximity of the large HAdV-C5 HI loop to CD46 (Fig. 8b). The residues involved in the predicted interaction are not conserved in any known CD46 interface and suggesting these are random interactions. Furthermore, interaction between the DG loop and CD46 is integral to known CD46 binding interfaces and is prevented by the HAdV-C5K HI loop laying between them (Fig. 8b).

**Biological evaluation of HAdV-D26/48K interaction with CD46.** Antibody competition inhibition assays in CHO-BC1 cells (CHO cells transduced to express the BC1 isoform of CD46) were used to test the predictions made by modelling (Fig. 8c). These data confirm that recombinant HAdV-D26K and HAdV-D48K proteins are incapable of inhibiting antibody binding to CD46 at

any concentration tested (up to 2 ng/cell), whilst the well-defined CD46 interacting HAdV-B35K demonstrates strong inhibition, with a calculated IC50 of 0.003 μg/$10^5$ cells.

SPR analysis of the interaction between recombinant fiber-knob protein with CD46 confirms these findings. The known CD46 utilising HAdV-B35K is seen to bind CD46 even at low concentration, while HAdV-D48K shows no interaction (Fig. 8d). HAdV-D26K shows a very low affinity interaction with CD46, however the kinetics are extremely fast making it impossible to measure an accurate $K_{On/Off}$ at any of the concentrations measured, suggesting an unstable interface. The calculated $K_D$ for HAdV-D26K is seen to be more than $1.5 \times 10^3$ times lower than that of HAdV-B35K (Fig. 8e).

**In silico evaluation of HAdV-D26/48K interaction with DSG-2.** The third major protein receptor for human adenoviruses is Desmoglein 2 (DSG2), shown to enable infection by HAdV-B3, B7, B11, and B14[43,44]. Whilst we have not been able to model the interaction of HAdV-D26 or D48 with DSG2, due to the lack of an available high-resolution complexed structure at the time of writing, we investigated the interaction by SPR analysis. HAdV-B3 is the best studied DSG2 binding adenovirus and showed binding in the μM range when tested by SPR (Fig. 9a), however, no binding was observed when the same experiment was run with HAdV-D26K or HAdV-D48K (Fig. 9b).

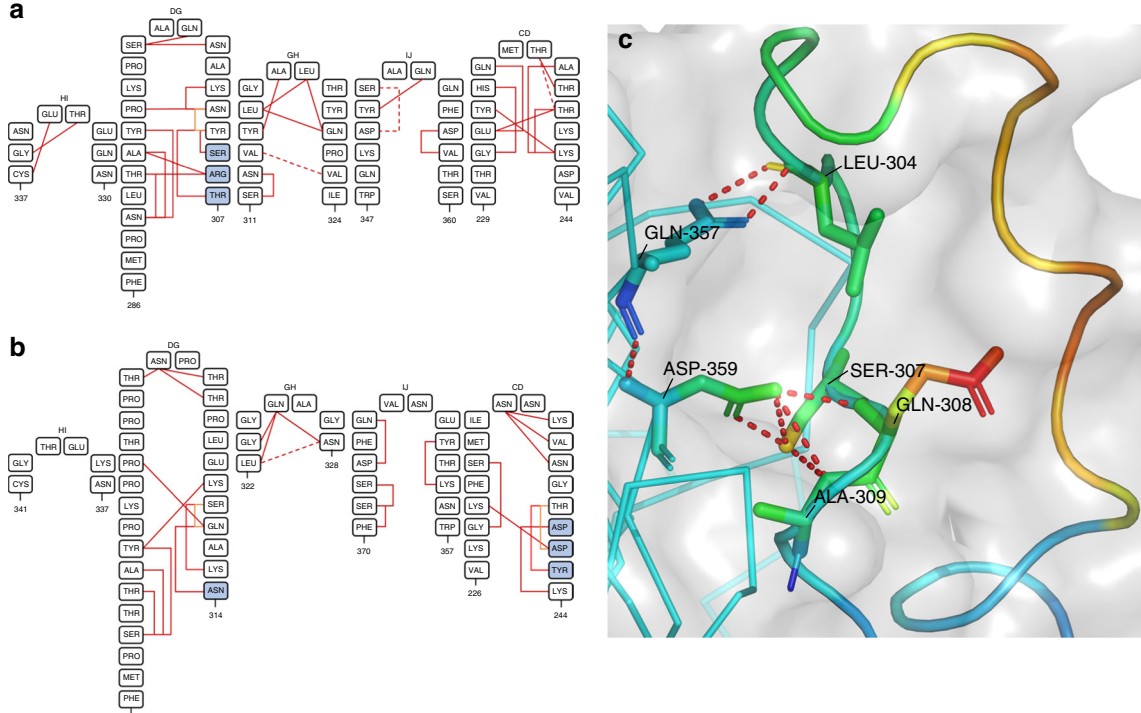

**Fig. 4** Hypervariable loop conformations and contacts residues. The residues comprising the indicated loops of HAdV-D26K (**a**) and HAdV-D48K (**b**) are shown diagrammatically with numbers indicating the start and end residues of each loop depicted. The network of intraloop polar interactions is shown by solid lines (one polar bond), and dashed lines (two polar bonds, colour variations are only for ease of viewing) between interacting residues, similar interloop bonds are also present as visualised in Supplementary Figure 2. Residues forming part of a helical motif are shaded in blue. The HAdV-D48K DG-loop is seen to form contacts to the opposing monomer across the inter-monomeric cleft (**c**). The labelled residues forming polar contacts (shown as sticks) are coloured by relative B-factor, with warmer colours indicating higher relative B-factors, as is the cartoon representation of the loop. The opposing HAdV-D48K monomer is seen as a ribbon representation of the carbon-α chain in cyan and the surface of the HAdV-D48K trimer seen as a semi-transparent grey surface

## Discussion

This study reveals the crystal structure of two adenovirus proteins critical to primary receptor engagement, HAdV-D26 and HAdV-D48 fiber-knob, which are important viral vectors currently in human clinical trials[3,11,12]. Despite their advanced development, the field lacks fundamental knowledge regarding the mechanisms of infection for these viral vector platforms. The work we described here provides a combined crystallographic, in silico, and in vitro approach to investigate adenovirus fiber-knob: receptor interactions with CAR and CD46, two receptors previously proposed to be utilised by these viruses[5,9,17].

Analysis of the phylogenetic relationship between 56 adenovirus serotypes, by both whole genome and fiber-knob domain alignment (Fig. 1a), confirms diversification into the widely accepted seven adenoviral species[7]. However, generating the phylogenetic tree with fiber-knob sequences, rather than whole genomes, shows additional diversity, not revealed by the whole virus taxonomy. Adenovirus species D breaks up into several additional sub-clades when focused on the fiber-knob, suggesting greater receptor diversity than might be expected based on the whole virus phylogeny. Similar observations have previously been made in species D hexon and penton[13].

In contrast to species D, the phylogeny of species B adenoviruses, which are known to utilise Desmoglein 2 and CD46 as primary receptors, is simplified when focused upon the fiber-knob, indicating less diverse receptor usage[39,43–45]. This simplification in comparison to the whole genomic alignment implies that much of the species diversity must lay in other proteins. The E3 protein, for example, is known to be highly diverse within species B adenoviruses, having previously been exploited in the selection of the oncolytic (cancer killing) virus enadenotucirev, which is currently in clinical trials[13,46,47].

That we see such opposing effects on the species B and D phylogenetic trees when focusing on the fiber-knob, highlights the limitations of simple taxonomic approaches. The current adenoviral taxonomy is based on antibody neutralisation assays, which are limited by antibody's reliance on surface accessible proteins in the capsid, and does not account for diversity in other viral proteins, as the above suggests for species B. This supports a taxonomic proposal based upon viral genetics rather than antibody neutralisation, as has previously been suggested[13,48].

Many studies on adenovirus neutralisation have focused upon NAbs which bind to the hexon[49–51]. Following intramuscular vaccination with non-replicating adenoviral vectors, most NAbs are targeted to the hexon; a reflection of its high abundance and surface availability in each viral capsid[49,52]. However, during natural infection, many NAbs target the fiber protein[52], presumably due to the abundance of fiber produced in the early stages of hAdV replication to loosen cell–cell junctions and facilitate viral spread, prior to lysis and entry of large amounts of whole virus to the bloodstream[53]. For individuals with pre-existing anti-adenovirus immunity derived from natural infections, anti-fiber NAbs are likely to limit the use of vectors with common adenovirus fiber proteins by neutralisation of the viral vector prior to its therapeutic effect. Thus, for the development of vectors to circumvent pre-existing anti-adenovirus immunity for therapeutic use further exploration of this fiber protein diversity

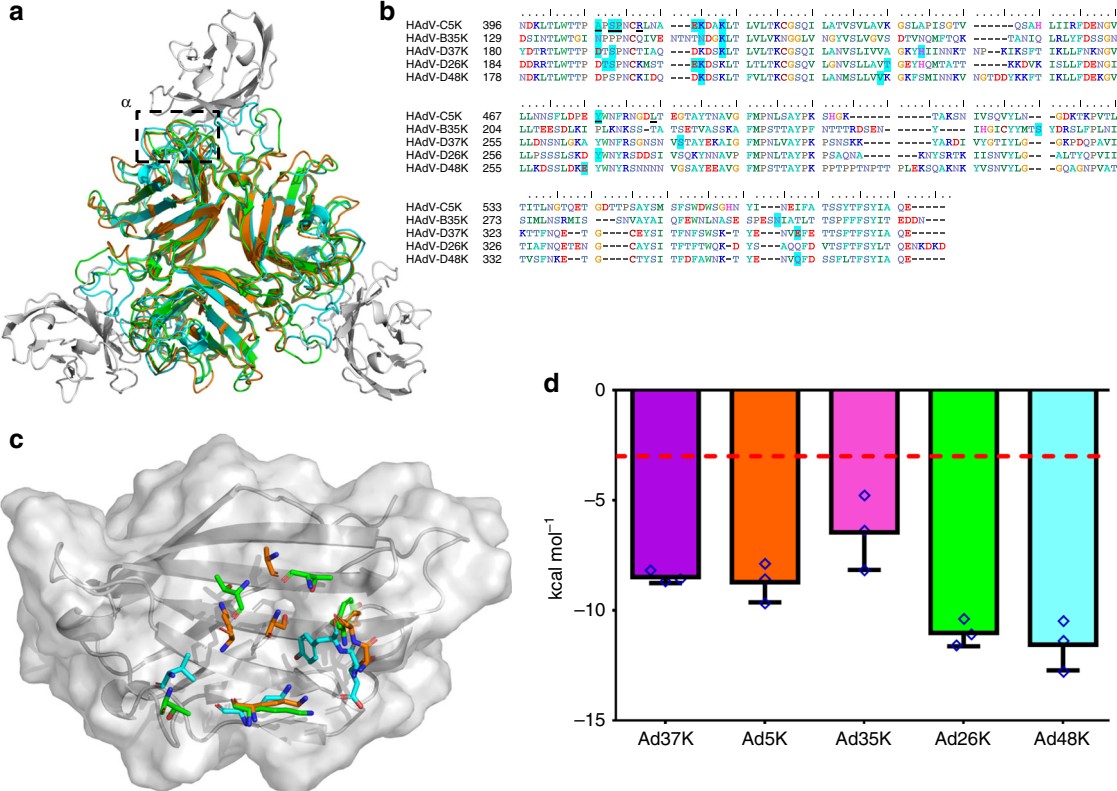

**Fig. 5** Modelling of the HAdV-D26K and HAdV-D48K interaction with CAR at the α-interface. The α-interface region is shown by the box on the structural alignment of HAdV-C5K (orange), 26K (green), and 48K (cyan) fiber-knob domain crystal structures in complex with CAR-D1 domain (grey) as determined by homology alignment to the previously reported Ad37K CAR-D1 structure (PDB: 2J12) (**a**). The aligned amino acid sequence of the investigated fiber-knobs (**b**) and the predicted α-interface forming CAR-D1 binding residues are highlighted in blue, with the underlined residues representing the HAdV-C5K amino acids shown by Kirby et al. (2000)[37] to be important for CAR interaction. Conservation of key residues can be seen between HAdV-C5K, HAdV-D26K, and HAdV-D48K fiber-knobs. This conservation is visualised, with the contact residues comprising the α-interface with HAdV-C5K, 26K, and 48K shown as sticks in complex with the energy minimised CAR-D1 domain (grey), shown as the surface of the maximum spatial occupancy of the aligned CAR-D1 monomers from each of the energy minimised models in complex with HAdV-C5K, 26K, and 48K fiber-knobs (**c**). **d** Plots the predicted binding energy of the energy minimised fiber-knob proteins to CAR-D1 complex in the α-interface, only. Lower binding energy indicates a more stable interface with the red line depicting 3.0 kcal mol$^{-1}$, which can be considered background. $n = 3$, where each calculation is an independent fiber-knob: CAR interface, error bars indicate mean ± SD

may be beneficial, as well as the on-going studies using hexon HVR pseudotypes to circumvent anti-hexon immunity[11,54].

Analysis of the adenovirus loops (Fig. 4, Supplementary Figure 3) reveals an intricate network of polar interactions which stabilise their three-dimensional structures. These bonds appear to hold the loops in a conformation which, in the case of HAdV-B11K and HAdV-B35K, facilitates receptor binding. In the HAdV-D26 and HAdV-D48 fiber-knob structures presented in this study the loops are also held in a stable conformation, though not one which enables CD46 interaction.

Modelling of HAdV-D26K/HAdV-D48K in complex with CD46 (Fig. 7) suggested few contacts, and interface energy calculations using these models predict a weak binding energy (Fig. 8). SPR indicated that HAdV-D26K has an affinity for CD46 that is approximately 1500× weaker than that of HAdV-B35K (Fig. 8d, e). Combined with the extremely fast kinetics, this is suggestive of a highly unstable interface. HAdV-D48K showed no affinity for CD46 at all. This was confirmed by in vitro competition inhibition assays, in which no tested quantity of recombinant fiber-knob was capable of inhibiting antibody binding to CD46 (Fig. 8c). These findings appear contradictory to previous studies which suggest CD46 as the primary receptor for these viruses[16,17]. Our findings improve knowledge of the cell entry mechanisms of these viruses and the vectors derived from them,

and do not diminish the observed effectiveness of these vaccines. However, if CD46, a protein expressed on the surface of all nucleated cells, is not the receptor for these viruses then it is as yet unknown what the primary tissue tropism determinant is for these clinically significant viruses[55,56].

A similar methodology was applied to the interaction HAdV-D26K/HAdV-D48K with CAR. Inspection of the modelled complexes (Fig. 5) indicated a conserved α-interface enabling CAR binding in adenovirus 5, 26, and 48, fiber knobs. However, the structure of the β-interface interaction appears to indicate a mechanism modulating the fiber-knob's CAR affinity (Fig. 6). When occupying the intermonomer cleft in the conformation shown in Fig. 6b, the DG loops of HAdV-D26K and HAdV-D48K are likely to inhibit CAR binding by steric hindrance, but if the loops were to shift into a conformation which relieves this clash CAR binding could occur. Therefore, the ability of these vectors to interact with CAR is likely a function of the steric hindrance provided by these loops, reducing the ability of the fiber-knob domain to engage CAR in a permissive conformation. SPR analysis supports this hypothesis (Fig. 6d, e), where the larger the DG loop of the investigated fiber-knob the slower the $K_{On}$.

The inter-loop contacts described in Fig. 4a, b, and Supplementary Figure 3, and the normalised B-factors described in Fig. 6b will influence the molecular dynamics of the DG-loops.

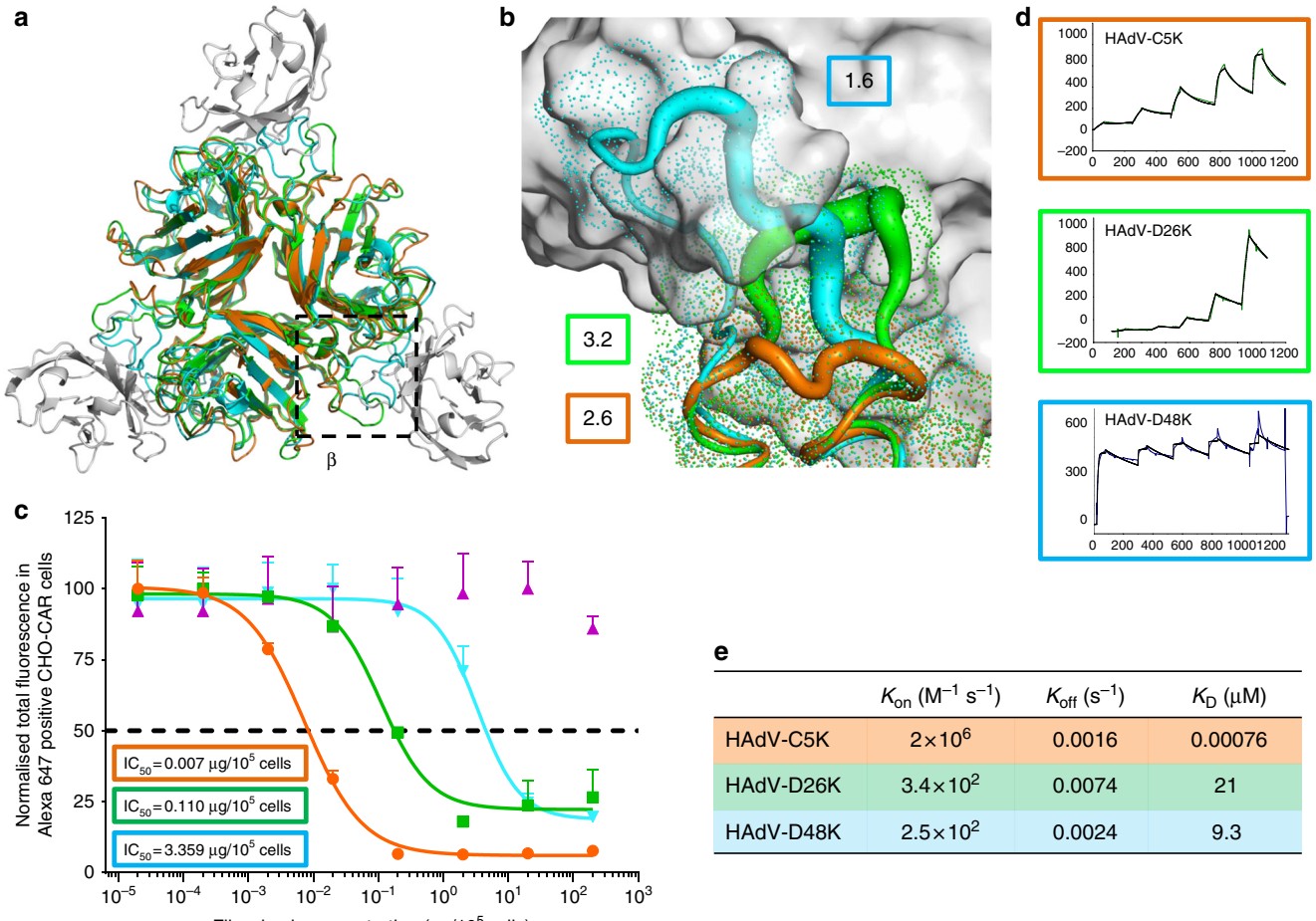

**Fig. 6** Modelling of the HAdV-D26K and HAdV-D48K interaction with CAR at the β-interface. The β-interface region is shown by the box on the structural alignment of HAdV-C5K (orange), 26K (green), and 48K (cyan) fiber-knob domain crystal structures in complex with CAR-D1 domain (grey) as determined by homology alignment to the previously reported Ad37K CAR-D1 structure (PDB: 2J12) (**a**). A dot surface shows the surface of HAdV-C5K, 26K, and 48K DG-loops in the inter-monomer cleft (**b**). The boxes denote the maximum B-factor of the corresponding loops, which are shown as putty representations with thicker regions indicating higher relative B-factors, from which we can infer the relative stability of the loops. Antibody competition inhibition assay (**c**) shows the relative inhibitory ability of the HAdV-C5, 35, 26, and 48, fiber-knob domains in CAR expressing CHO-CAR cells, with the calculated $IC_{50}$ values shown in boxes. $n = 3$ biological replicates. Surface plasmon resonance (SPR) traces are shown by coloured lines, and the fitted curves by black lines (**d**). The calculated binding coefficients on rate ($K_{On}$), off rate ($K_{Off}$), and dissociation coefficient ($K_D$) are given in the table (**e**). $IC_{50}$ curves are fitted by non-linear regression. Error bars represent standard deviation of 3 biological replicates. Error bars indicate mean ± SD

Loops which can occupy a CAR inhibitory conformation but have fewer stabilising contacts, such as that of HAdV-D26K (Supplementary Figure 3A, C), should be more permissive to CAR binding. While loops which are less flexible and/or stabilised in a CAR inhibitory conformation, such as HAdV-D48K (Fig. 6b, Supplementary Figure 3B, D) should result in a fiber-knob which is less able to bind CAR. This hypothesis fits the competition inhibition studies shown in Fig. 6c, which demonstrate that HAdV-D26K has an approximately ~15× lower affinity for CAR than HAdV-C5K, and HAdV-D48K has 500× lower affinity.

Interestingly, the affinity of HAdV-D48K for CAR as measured by SPR is approximately 2× higher than that for HAdV-D26K, due to the slower $K_{Off}$ of HAdV-D48K (Fig. 6e) which is in contrast with the $IC_{50}$ curves (Fig. 6c) in which HAdV-D48K is observed to bind to CAR less strongly than HAdV-D26K. The incongruity may be explained by the methodology. It is possible that the large fluid volume in the wells during the inhibition assay (in comparison to the BIAcore microfluidics system), favoured greater binding by HAdV-D26K due to its faster $K_{On}$, compared to HAdV-D48K. This discrepancy does not alter the proposed model of CAR interaction, and seems to confirm the importance of the $K_{On}$, presumably mediated by the β-interface.

Species D adenoviruses have a large range of different DG loops (Supplementary Figure 5). Most sequences have lengths equal to, or greater than, that of HAdV-D26K, making it plausible that they too could modulate the fiber-knob interaction with CAR. However, the magnitude of this effect will be dependent on the individual molecular dynamics of the DG-loops and its interactions with adjacent residues.

Assuming this mechanism of CAR binding regulation is broadly applicable, it may have important implications for adenoviral vector design. The presence of a high affinity receptor for the virus can mask the low affinity CAR interaction, creating a hidden tropism only observed if the virus is forced to rely upon it. Expression of CAR on human erythrocytes suggests the potential for sequestration of virotherapies in the blood[57]. CAR expression in lung epithelial tissues offers another site for potential off target activity[58,59]. Therefore, many virotherapies previously thought to be non-CAR binding adenoviruses may in fact demonstrate weak CAR tropism, driving off target infections or resulting in

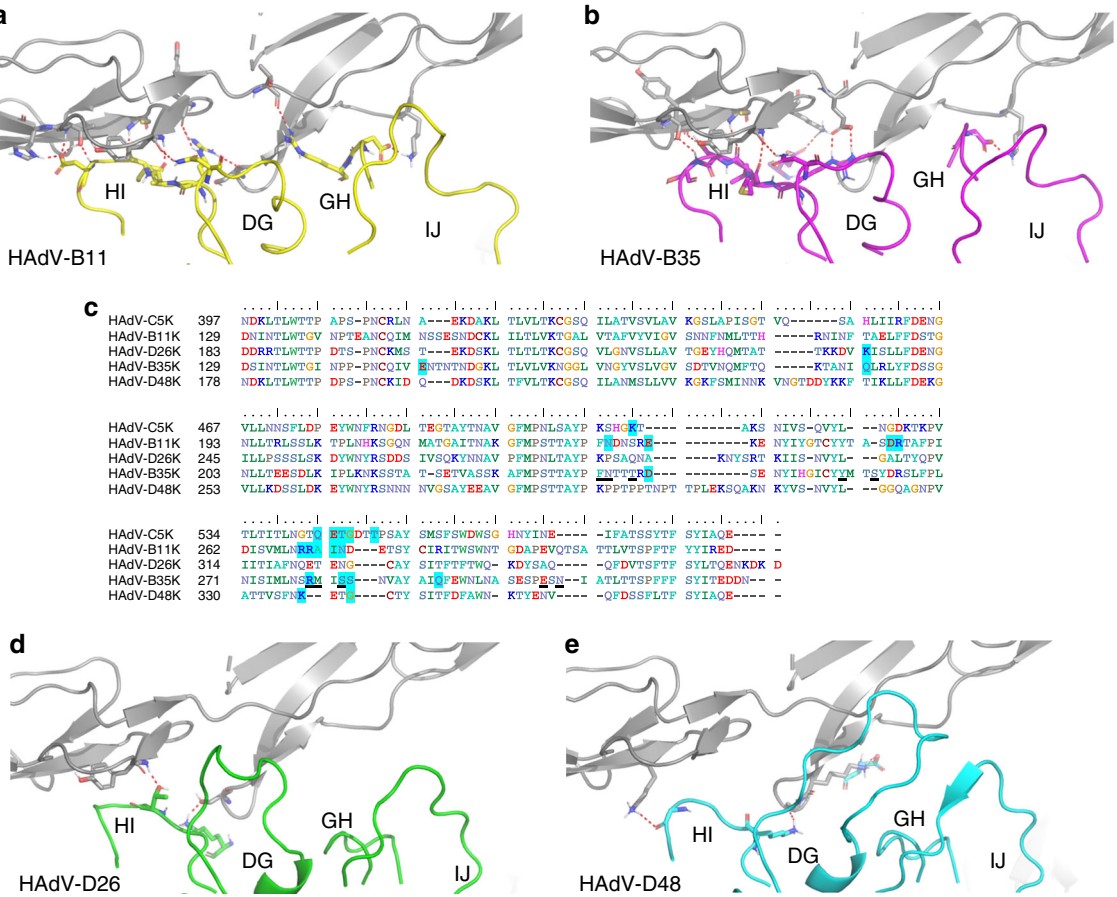

**Fig. 7** Modelling of the HAdV-D26K and HAdV-D48K with CD46. Red dashes show contacts between the energy minimised crystal structure of CD46 SC1 and SC2 domains (grey cartoon) and Ad11K in complex (PDB 3O8E) (**a**). The known CD46 interacting fiber-knob, HAdV-B35K (purple), is aligned to the above crystal structure and energy minimised (**b**). Amino acid sequence alignment of the tested fiber-knob proteins (**c**) shows conservation of residues previously shown by Wang et al. (2007)[36] to be key to CD46 binding (underlined) between the known CD46 binding fiber-knobs, Ad11K and HAdV-B35K. Residues highlighted in blue are predicted to form direct contacts with CD46 in the energy minimised models. Similar alignments to that performed with HAdV-B35K are shown for HAdV-D26K (Green—**d**) and HAdV-D48K (cyan—**e**). In all models, red dashes indicate polar contacts between the residues shown as stick representations

sequestration of the vector in tissues other than that target. This may not be of grave consequence for non-replicating vectors, such as viral vaccines, but in vectors which rely upon controlled replication in targeted tissues, such as oncolytic virotherapies, this could result in off-target infection, dysregulated expression of therapeutic protein, and reduced delivery to the point of need.

DSG2 was also shown to be unable to bind HAdV-D26K or HAdV-D48K at any concentration by SPR. It is notable that the $K_D$ measured for the HAdV-B3K (66.9 μM) is much lower than that measured during the original investigation of DSG2 as an HAdV-B3K receptor (2.3 nM)[44]. This is likely due to our use of recombinant knob trimers, rather than the multivalent penton dodecahedrons.

The final, known, adenovirus fiber-knob receptor, which has thus far not been addressed in this study is sialic acid, as part of glycosylation motifs. Several adenoviruses have been shown to bind to sialylated glycans, including HAdV-D37[32,38], HAdV-G52K[30,60], and Canine adenovirus serotype 2 (CAdV-2)[57]. Each of these three viruses binds to sialic acid by different mechanisms (Supplementary Figure 6). Supplementary Figure 6 shows that HAdV-D26/48K do not share the sialic acid binding residues found in HAdV-G52K or CAV-2 but do share the Tyr-142 and Lys-178 contact residues with HAdV-D37K. Further, the HAdV-D37K contact residue Pro-147 is between the sialic acid and the main chain oxygen which is functionally identical at the similar position in HAdV-D26/48K. Taken

together, it remains plausible that HAdV-D26/48K may be capable of binding sialic acid in an HAdV-D37K-like manner. However, binding does not equate to functional infection, as seen with HAdV-D19pK[32] and further studies are required to ascertain whether HAdV-D26/48 are capable of utilising sialic acid to generate a productive infection. Further, HAdV-D37 was shown to require a specific glycosylation motif (GD1a) in order form a functional infection, so any assessment of sialic acid as an adenoviral receptor must be in the context of its glycan carrier[38].

The work undertaken in this study presents, for the first time, the crystal structures of the fiber-knob domain protein of HAdV-D26 (PDB: 6FJN), and HAdV-D48 (PDB: 6FJQ) fiber-knob protein. In addition, we report a new crystal structure for HAdV-C5 fiber-knob protein (PDB: 6HCN) with improved resolution compared to the existing structure (PDB: 1KNB)[27]. We utilised these structures to investigate the ability of these proteins to interact with the putative receptors, CAR and CD46, by an integrative structural, in silico, and in vitro workflow. We demonstrate that HAdV-D26 and HAdV-D48 fiber-knob domains have a weak ability to bind CAR, and negligible CD46 interaction, suggesting that these viruses are unlikely to utilise these proteins as a primary receptor in vivo. Finally, we showed that DSG2 is also unable to form a stable interaction in the context of SPR analysis. We suggest that CAR binding is moderated by a previously unreported mechanism of steric inhibition

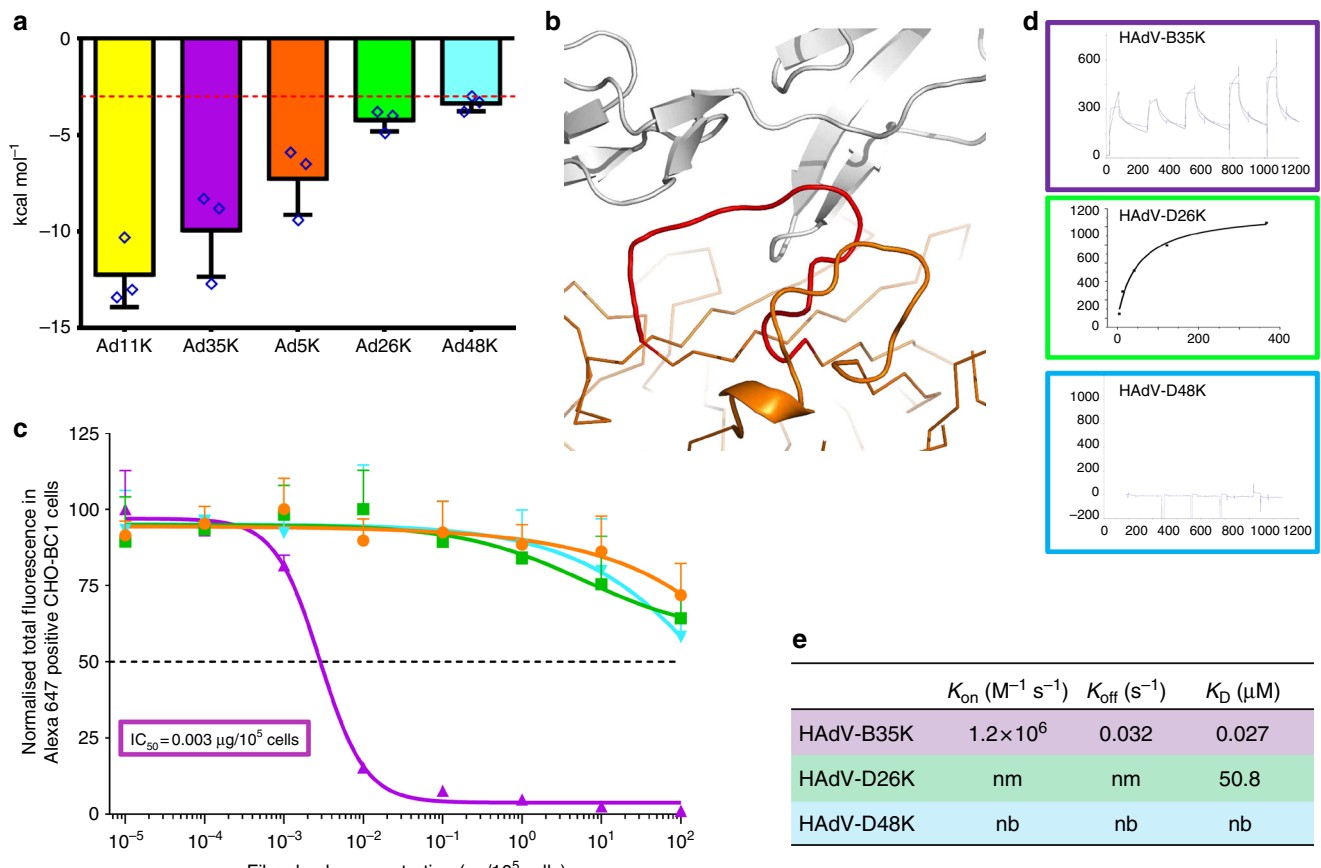

**Fig. 8** Binding energetics and affinities of HAdV-D26K and HAdV-D48K with CD46. Calculation of the predicted binding energies for the energy minimised fiber-knob: CD46 models are compared on the bar chart (**a**), lower kcal mol$^{-1}$ values indicate a stronger interaction, the red line at 3.0 kcal mol$^{-1}$, denotes an interface energy which can be considered negligible (random proteins passing in solution), $n = 3$, where each calculation is an independent fiber-knob: CD46 interface. The HI loop (red) of the HAdV-C5 fiber-knob (orange) is seen to extend between CD46 (grey) and the DG loop (**b**). The antibody competition inhibition assay (**c**) shows the relative inhibitory ability of the HAdV-C5, HAdV-B35, HAdV-D26, and HAdV-D48 fiber-knob domains in CD46 expressing CHO-BC1 cells, with the calculated IC$_{50}$ values shown in boxes. $n = 3$ biological replicates. Surface plasmon resonance (SPR) traces are shown by coloured lines, and the fitted curves by black lines (**d**). The calculated binding coefficients on rate ($K_{On}$), off rate ($K_{Off}$), and dissociation coefficient ($K_D$) are given in the table (**e**), nm (not measured) indicates that the kinetics were too fast to measure, nb denotes no binding. IC$_{50}$ curves are fitted by non-linear regression. Error bars represent standard deviation of 3 biological replicates. Error bars indicate mean ± SD

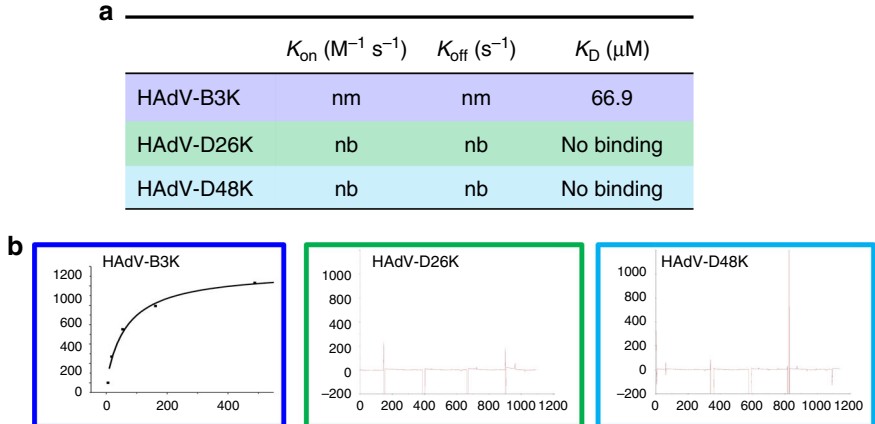

**Fig. 9** Desmoglein 2 is unlikely to be a receptor for HAdV-D26K or HAdV-D48K. The dissociation constant was calculated for HAdV-B3K binding to DSG2, but kinetics were too fast to determine $K_{On}$ or $K_{Off}$ (**a**), the $K_D$ curve is shown for HAdV-B3K while HAdV-D26K and HAdV-D48K are seen to form no interaction with DSG2 (**b**). nm (not measured) indicates that the kinetics were too fast to measure, nb denotes no binding

which may apply to other adenoviruses and demonstrate an in silico methodology capable of rapidly predicting these interactions. These findings enhance our understanding of the virology of adenovirus infection, and have direct implications for virotherapy vector design, which often rely upon carefully controlled receptor tropisms to achieve specificity and efficacy[9,18,28].

## Methods

**Genome alignment and analysis of genetic diversity.** Representative whole genomes (nucleotide) of adenoviral species 1–56 were selected from the National Center for Biotechnology Information (NCBI), and aligned using the EMBL-EBI Clustal Omega tool[61,62]. Fiber-knob domain amino acid sequences were derived from the same genome sequences, defined as the translated nucleotide sequence of the fiber protein (pIV) from the conserved TLW hinge motif to the protein C-terminus, and aligned in the same manner as the whole genomes. Phylogenetic relationships were inferred using the maximum likelihood method based upon the Jukes Cantor model for the whole genome nucleotide analysis[63], and the Poisson correction model for the fiber-knob amino acid analysis[64], using MEGA X software[65]. Confidence was determined by bootstrap analysis (500 replicates)[66] and trees displayed condensed at 70% confidence (percentage confidence values shown at each node) where stated.

**Fiber-knob amino acid variability.** Amino acid sequence variability scores were calculated from the Clustal Omega aligned amino acid sequences of the fiber-knob domains of either adenoviruses 1–56, or only the species D adenoviruses. Analysis was performed using the Protein Variability Server (PVS), using a consensus base sequence and the Wu–Kabat method[67].

**Generation of recombinant fiber-knob protein.** SG13009 *Escherichia coli* har bouring pREP-4 plasmid and pQE-30 expression vector containing the fiber-knob DNA sequence were cultured in 20 ml LB broth with 100 μg/ml ampicillin and 50 μg/ml kanamycin overnight from glycerol stocks made in previous studies[18,68,69]. 1 L of TB (Terrific Broth, modified, Sigma-Aldrich) containing 100 μg/ml ampicillin and 50 μg/ml were inoculated with the overnight *E. coli* culture and incubated at 37 °C until they reached OD0.6. IPTG was then added to a final concentration of 0.5 mM and the culture incubated at 37 °C for 4 h. Cells were then harvested by centrifugation at 3000*g*, resuspended in lysis buffer (50 mM Tris, pH 8.0, 300 mM NaCl, 1% (v/v) NP40, 1 mg/ml Lysozyme, 1 mM β-mercaptoethanol), and incubated at room temperature for 30 min. Lysate was clarified by centrifugation at 30,000*g* for 30 min and filtered through a 0.22 μm syringe filter (Millipore, Abingdon, UK). Clarified lysate was then loaded onto a 5 ml HisTrap FF nickel affinity chromatography column (GE) at 2.0 ml/min and washed with 5 column volumes into elution buffer A (50 mM Tris [pH 8.0], 300 mM NaCl, 1 mM β-mercaptoethanol). Protein was eluted by 30 min gradient elution from buffer A to B (buffer A + 400 mM Imidazole). Fractions were analysed by reducing SDS-PAGE, and fiber-knob containing fractions further purified using a superdex 200 10/300 size exclusion chromatography column (GE) in crystallisation buffer (10 mM Tris [pH 8.0] and 30 mM NaCl). Fractions were analysed by SDS-PAGE and pure fractions concentrated by centrifugation in Vivaspin 10,000 MWCO (Sartorius, Goettingen, Germany) proceeding crystallisation.

**Competition inhibition assays.** CHO cells expressing the appropriate receptor (CAR: CHO-CAR, or CD46: CHO-BC1) were seeded at a density of 30,000 cells per well in a flat bottomed 96-well tissue culture plate and incubated at 37 °C overnight. Serial dilutions were made up in serum-free RPMI-1640 to give a final concentration range of 0.0001–100 μg/10⁵ cells of recombinant soluble knob protein. Cells were incubated on ice for 30 min, then washed twice with cold PBS. Fiber-knob dilutions were then added to the cells and incubated on ice for 30 min. Cells were then washed twice in cold PBS and stained with the primary CAR or CD46 antibody, RmcB (Millipore; 05-644) or MEM-258 (Abcam; Ab789), respectively, to complex receptors unbound by fiber-knob protein, and incubated for 1 h on ice. Cells were washed twice further in PBS and incubated on ice for 1 h with Alexa-647 labelled goat anti-mouse F(ab')2 (ThermoFisher; A-21237)[18,68,69]. All antibodies were used at a concentration of 2 μg/ml.

Samples were run in triplicate and analysed by flow cytometry on Attune NxT (ThermoFisher), and analysed using FlowJo v10 (FlowJo, LLC) by gating sequentially on singlets, cell population, and Alexa-647 positive cells. Total fluorescence (TF) was used as the measure of inhibition, rather than percentage of fluorescent cells in the total population, to account for the presence of multiple receptor copies per cell surface which can enable partial inhibition of antibody binding on the cell surface. TF was defined as the percentage of Alexa-647 positive cells in the single cell population for each sample and multiplied by the median fluorescent intensity (MFI) of the Alexa-647 positive single cell population in each sample. Data points are the mean total fluorescence of three biological replicates with error given as the standard deviation from the mean. IC₅₀ curves were fitted by non-linear regression, and used to determine the IC₅₀ concentrations[18,68,69]. CHO-CAR and CHO-BC1 cells were originally derived by Bergelson et al.[70] and Manchester et al.[71], respectively.

**Crystallisation and structure determination.** Protein samples were purified into crystallisation buffer (10 mM Tris [pH 8.0] and 30 mM NaCl). The final protein concentration was approximately 7.5 mg/ml. Commercial crystallisation screen solutions were dispensed into 96-well plates using an Art-Robbins Instruments Griffon dispensing robot (Alpha Biotech, Ltd.), in sitting-drop vapour-diffusion format. Drops containing 200 nl of screen solution and 200 nl of protein solution were equilibrated against a reservoir of 60 μl crystallisation solution. The plates were sealed and incubated at 18 °C.

Crystals of HAdV-C5K appeared in PACT *Premier* condition D04 (0.1 M MMT, pH 7.0, 20% PEG 1500), within 1–7 days. Crystals of HAdV-D26K appeared within 1–7 days, in PACT *Premier* (Molecular Dimensions, Suffolk, UK) condition A04; 0.1 M MMT [DL-malic acid, MES monohydrate, Tris], pH 6.0, 25% PEG 1500. Crystals of HAdV-D48K appeared in PACT *Premier* condition D02 (0.1 M Bis–Tris-propane, pH 6.5, 20% PEG 3350, 0.2 M NaNO₃), within 2 weeks. Crystals were cryoprotected with reservoir solution to which ethylene glycol was added at a final concentration of 25%. Crystals were harvested in thin plastic loops and stored in liquid nitrogen for transfer to the synchrotron. Data were collected at Diamond Light Source beamline I04, running at a wavelength of 0.9795 Å. During data collection, crystals were maintained in a cold air stream at 100 K. Dectris Pilatus 6M detectors recorded the diffraction patterns, which were analysed and reduced with XDS, Xia2[72], DIALS, and Autoproc[73]. Scaling and merging data was completed with Pointless, Aimless and Truncate from the CCP4 package[74]. Structures were solved with PHASER[75], COOT[76] was used to correct the sequences and adjust the models, REFMAC5[77] was used to refine the structures and calculate maps. Graphical representations were prepared with PyMOL[78]. Reflection data and final models were deposited in the PDB database with accession codes: HAdV-C5K, 6HCN; HAdV-D26k, 6FJN; and HAdV-D48k, 6JFQ. Full crystallographic refinement statistics are given in Supplementary Table 2; stereo images depicting representative areas of the model and map are provided in Supplementary Figure 7.

**Modelling of fiber-knob ligand interactions.** Fiber-knob proteins were modelled in complex with CAR or CD46 using the existing HAdV-D37K—CAR liganded (PDB 2J12) or the HAdV-B11K—CD46 liganded (PDB 3O8E) structures, respectively, as a template. Non-protein components and hydrogens were removed from the template model and the fiber-knob protein of interest. The two fiber-knob proteins were then aligned with respect to their Cα chains, in such a way as to achieve the lowest possible RMSD. Models containing only the fiber-knob protein of interest and the ligand were saved and subjected to energy minimisation, using the YASARA self-parametrising energy minimisation algorithm as performed by the YASARA energy minimisation server, and results were visualised in PyMol[78,79].

**Calculation of interface energy.** Interface energies were calculated using QT-PISA using biological protein assemblies and excluding crystallographic interfaces[80]. Values are the mean of the three symmetrical interfaces in each trimer and error is the standard deviation from the mean, any values above −3.0 kcal mol⁻¹ were considered to be background as shown as a red dashed line on graphs[42].

**Sequence alignments.** Alignments were performed using the Clustal Omega multiple sequence alignment algorithm and visualised with BioEdit[61,62].

**B-factor normalisation.** Comparing order between different structures by comparing individual B-factors can be misleading. Post-refinement B-factors relate to the Wilson B-factor, which can vary widely between data sets, even from the same crystal preparation. A valid comparison between different structures can be achieved by comparing normalised B-factors instead. Normalisation was performed by dividing individual atomic B-factors by the average B-factor of the whole data set, quantifying the range of internal flexibility in a structure. This normalised B-factor can then be compared between different data sets.

**Surface plasmon resonance (SPR) analysis.** Binding analysis was performed using a BIAcore 3000™ equipped with a CM5 sensor chip. Approximately 5000 RU of CD46, CAR, and DSG2 was attached to the CM5 sensor chip, using amine coupling, at a slow flow-rate of 10 μl/min to ensure uniform distribution on the chip surface. A blank flow cell was used as a negative control surface on flow cell 1. All measurements were performed at 25 °C in PBS buffer (Sigma, UK) at a flow rate of 30 μl/min. For equilibrium binding analysis, the HAdV-D26K and HAdV-B3K fiber knob proteins were purified and concentrated to 367 and 3 μM respectively. 5× 1:3 serial dilutions were prepared for each sample and injected over the relevant sensor chip. The equilibrium binding constant ($K_D$) values were calculated assuming a 1:1 interaction by plotting specific equilibrium-binding responses against protein concentrations followed by non-linear least squares fitting of the Langmuir binding equation. For single cycle kinetic analysis, HAdV-D26K, HAdV-D48K, HAdV-B35K, HAdV-C5K, and HAdV-B3K were injected at a top concentration of around 200 μM, followed by four injections using serial 1:3 dilutions. The $K_D$ values were calculated assuming Langmuir binding (AB = B×ABmax/(KD + B)) and the data were analysed using the kinetic titration algorithm (BIAevaluationTM 3.1). Receptor proteins were obtained commercially, as follows: Recombinant Human Desmoglein-2 Fc Chimera Protein, R&D Systems, Catalogue number 947-DM-100. Recombinant Human CXADR Fc Chimera Protein (CAR),

R&D Systems, Catalogue number 3336-CX-050. Recombinant Human CD46 Protein (His Tag), Sino Biological, Catalogue number 12239-H08H.

**Reporting summary**. Further information on experimental design is available in the Nature Research Reporting Summary linked to this article.

## Data availability

Macromolecular structures generated during this study have been deposited in wwPDB (worldwide Protein Data Bank; https://www.wwpdb.org/), and have PDB ID's 6FJN, 6HCN, and 6FJQ. PDB ID's for macromolecular structures utilised, but not generated in the course of this study, are as follows: HAdV-B11K in complex with CD46, PDB 3O8E. HAdV-D37K in complex with CAR-D1, PDB 2J12. HAdV-B35K, PDB 2QLK. Genomic sequences from which fiber-knob domain sequences were determined, which have been used in phylogenetic analysis, have the following NCBI accession numbers: AC_000017| AF532578|X73487|AY803294|AB562586|AY601636|AF108105|GU191019|JQ326209| AC_000007|JN226749|KF528688|FJ404771|JN226750|JN226751|JN226752|EF153474| JN226753|FJ824826|JN226754|JN226755|AM749299|JN226756|JN226758|AY737797| AC_000019|GQ384080|JN226759|JN226760|KU162869|DQ315364|JN226761|JN226762| JN226763|JN226764|AY875648|JN226757|EF153473|DQ393829|AC_000008|AY737798| JN226765|DQ923122|AB605243|NC_012959|FJ643676|HM770721|HQ413315| AC_000018|DQ086466|JN226746|JN226747|AB448776|AB448767|AJ854486|KF006344|. All other data pertaining to this manuscript are available from the authors upon request.

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

## Acknowledgements

A.T.B. is supported by a Tenovus Cancer Care PhD studentship to A.L.P. (reference PhD2015/L13). A.G.-W. was supported by a Life Sciences Research Network Wales (LSRNW) PhD studentship. J.A.D. is supported by a Cancer Research UK Biotherapeutics Drug Discovery Project Award to A.L.P. (project reference C52915/A23946). H.U.-K. was supported by a Cancer Research Wales PhD studentship to A.L.P. L.C. is funded by The HC Roscoe Grant 2016 from the British Medical Association Foundation for Medical Research and by the National Institutes for Health (NIH)/National Institute of Allergy and Infectious Diseases (NIAID) under CEIRS contract HHSN272201400008C. A.L.P. and P.J.R. are funded by Higher Education Funding Council for Wales. The authors acknowledge the Diamond Light Source for beamtime (proposal mx14843) and the staff of beamline I04 for assistance with diffraction data collection. The authors acknowledge Johanne Pentier for technical assistance with SPR data collection, as well as Aaron Wall and Anna Fuller for assistance with FPLC. The authors also acknowledge Elmar Kreiger and the team who maintain the YASARA energy minimisation server, and Pedro Reche and the Immunomedicine Group for maintenance of the Protein Variability Server.

## Author contributions

A.T.B. and A.L.P. conceived and designed the study. Modelling of protein–protein interfaces, phylogenetics, protein variability, and interface energy calculations were performed by A.T.B. A.T.B. and A.G.-W. performed crystallisation experiments. A.T.B., A.G.-W., and P.J.R. solved and refined crystallographic structures, and analysed the resultant models. Competition inhibition studies were performed by A.T.B. with advice from L.C. A.T.B. and D.K.C. performed SPR experiments. H.U.-K., J.A.D., and L.C. provided DNA constructs and preliminary data. The manuscript was prepared by A.T.B. and A.L.P.; all other authors reviewed, edited, and approved the manuscript. The study was supervised by A.L.P.

## Additional information

**Competing interests:** The authors declare no competing interests.

