## [Peer Review File · Nature Communications]

Reviewers' Comments:

Reviewer #1:

Remarks to the Author:

The manuscript entitled "Diversity within the Adenovirus fiber knob hypervariable loops influences primary receptor interactions" by Baker et al., reports primarily the crystal structures of the fiber knobs of serotype-D adenoviruses Ad26 and Ad48. Even though the manuscript includes phylogenetic analysis, binding/inhibition assays and structure-based modeling studies, in addition to crystal structure determinations, there is nothing particularly novel in this report. However, if it is chosen for publication in Nature communications, the following queries need to be addressed.

1) The way the authors have chosen to represent the phylogenetic (circular) dendrograms is not quite meaningful. It appears that ends of branches appear to be equidistance for every pair within each diagram, whether it is of knob molecules or whole genome sequences, which is not quite informative. The conventional linear dedrograms would be meaningful.

2) Significantly, the authors primarily considered the Ad26K and Ad48K of species D Ads, binding to CAR and CD46. However, the species D fiber knobs are commonly known to bind sialic acid moieties as part of the larger complex carbohydrate receptors on the cell surfaces. It is puzzling, why the authors did not investigate the above fiber knobs binding sialic acid moieties or didn't even discuss such a possibility. This appears to a significant oversight.

3) The computational calculations of binding energies between the fiber knob and receptor molecules is not convincing. For instance, it appears that there are a number of steric clashes in the complex between the CAR and Ad26K (green) and Ad48K (cyan), as the loops running into the CAR surface (Fig. 6B). If there are indeed such steric clashes, then the bar graphs shown in the Figures 5D and 8A are not meaningful. Additionally, based on the Fig. 5D, there appear to be stronger or comparable interactions between CAR and Ad26K/Ad48K in comparison to Ad5K (Fig. 5D), which is not convincing and supports my point of view that there could be some steric clashes.

Reviewer #2:

Remarks to the Author:

The manuscript by Baker et al. presents two new human adenovirus type fibre head structures, both from the D family. Both these adenoviruses are therapeutically important. The structural work appears to be performed well, and the structures are analysed in detail. The structures suggest low or no binding to the common human adenovirus receptors CAR and CD46, which is confirmed experimentally. In the case of HAdV-D26, these results strongly contradict those in reference 17. Ideally, this should be further investigated, but the information provided here is already important for the field, even if the real receptors have not been identified. Some human and non-human adenoviruses have been shown to bind carbohydrate receptors, and this could perhaps also be investigated, to start off, by a glycan array, for example. In my opinion, the manuscript is already very interesting to the adenovirus community and has implications for a wider therapeutical field. As ever, some additional experiments could make it even more interesting, but I feel it is up to the Editors to decide if these would be necessary, because the current version is already internally consistent (i.e. no unsupported claims are made).

I also have the following, minor, suggestions to improve the manuscript:

- Indicate more clearly that the paper is about human adenoviruses.

- In a general journal like Nature Communications, a brief introduction to overall adenovirus structure and the fibre structure with a figure (spliced into Fig 1 if necessary), would be welcome.
- I personally don't like abbreviating adenovirus to Ad. Could be written in full each time, it is not a long word. HAdV, HAdV-C5 etc. is also better, according to current taxonomic standards. The same for NAbs, perhaps better written in full each time. But the Editors could say what they prefer.
- The supplemental crystallographic table has a missing parenthesis and lines with different fonts and sizes.
- When discussing the stability of loops (e.g. around line 136), possible stabilisation by crystal packing interactions should be analysed and discussed.

52: increaseD

63: delete comma?

220: interactionS

277-278: needs citations (i.e. repeat those given earlier)

410: PyMoll should be PyMol

416-418: can be written as a sentence.

Reviewer #3:

Remarks to the Author:

The work presented by Baker et al. describes the structure of the HAd26 and HAd48 fibre knobs and compares them from a phylogenetic and structural point of view with fibre knobs of other adenovirus serotypes. These two adenovirus serotypes represent a major interest in therapy due to their developments in clinical trials in the vaccination field against Ebola or HIV. Modelling approaches for interaction with two out of the three main protein receptors CAR and CD46 (the third one DSG2 being simply mentioned since the structure of the complex is not yet available).

Overall, the work is well presented and brings important new elements concerning the interaction of HAd26K and HAd48K with CAR. The length and rigidity of the DG loop of the fibre would be responsible for an interaction of 15 to 500 times less important than for the HAd5 used as a reference in this study. Rare points of contact would be possible with CD46, suggesting that other receptors may be needed for these two serotypes, which is important for understanding the tropism of these vectors of therapeutic interest.

- Although the competition data is convincing, it would have been interesting to have direct interaction data between the heads and CAR and CD46.
- A discussion stating if DSG2 is a rejected hypothesis for HAd26 and HAd48 would mean that new receptors remain to be identified. This point would be reinforced in the discussion section.

Response to reviewers: NCOMMS-18-26060-T:

Reviewer 1:

The manuscript entitled “Diversity within the Adenovirus fiber knob hypervariable loops influences primary receptor interactions” by Baker et al., reports primarily the crystal structures of the fiber knobs of serotype-D adenoviruses Ad26 and Ad48. Even though the manuscript includes phylogenetic analysis, binding/inhibition assays and structure-based modeling studies, in addition to crystal structure determinations, there is nothing particularly novel in this report. However, if it is chosen for publication in Nature communications, the following queries need to be addressed.

Author response: We thank the reviewer for their precis and assessment of our manuscript. We are disappointed to hear that s/he considers the data reported to be of limited novelty. We obviously, but respectfully, disagree. As highlighted in the manuscript, the vectors described are under late stage evaluation as therapeutic vaccine platforms, and therefore new insights into cellular interactions and mechanisms of action are extremely valuable from both a scientific, and clinical standpoint. While there is limited novelty in the nature of the experimentation as all techniques have been utilised previously, “novel crystal structures” are by their very definition, novel. With regards to the inhibition assays, whilst the technique is tried and tested, the result is important as it addresses what is a clear misconception/point of confusion in the field which has important implications for therapeutic vector development. The structure-based modelling studies are a new application of both existing data and newly determined structures, representing an entirely new analytical methodology which can be immediately deployed across our field.

1) The way the authors have chosen to represent the phylogenetic (circular) dendrograms is not quite meaningful. It appears that ends of branches appear to be equidistance for every pair within each diagram, whether it is of knob molecules or whole genome sequences, which is not quite informative. The conventional linear dedrograms would be meaningful.

Author response: We thank the reviewer for this comment, and now provide the requested additional, conventional dendrograms as a supplemental figure (Supplemental Figure 1). We retain our existing dendrograms since this is not intended to enable inference of ancestral relationships, but to provide a holistic view of the diversity in adenoviruses on a receptor only level with the whole genome alignment for comparison. We have shown a compressed tree (70% confidence) so as not to overestimate the diversity by including poorly supported nodes (defined here as <70% bootstrap confidence). To fully address the reviewer’s comment, we also include the following statement to

highlight the new dendrograms as a supplemental figure, and to outline our rationale for retaining the existing dendrograms (see lines 88-90)

“These phylogenetic trees have been condensed to 70% bootstrap confidence to exclude poorly supported nodes and display the projected diversity. A full dendrogram showing to-scale branches is provided in Supplemental figure 1.”

2) Significantly, the authors primarily considered the Ad26K and Ad48K of species D Ads, binding to CAR and CD46. However, the species D fiber knobs are commonly known to bind sialic acid moieties as part of the larger complex carbohydrate receptors on the cell surfaces. It is puzzling, why the authors did not investigate the above fiber knobs binding sialic acid moieties or didn't even discuss such a possibility. This appears to a significant oversight.

Author response: *We thank the reviewer for this comment. Our paper was intended to address the misconceptions surrounding CD46 interactions, which have dogged the literature for some time, whilst providing new insight into the mechanism of CAR interaction, explaining the virus's kinetics when exposed to CAR. We did not investigate sialic acid as there has been no previous suggestion that these viruses bind sialic acids, and crucially, there are no available models by which to model their mechanisms of interaction (the sialic acid complex models do not include the whole glycan so cannot be used in the same way). Whilst we consider that a robust analysis of sialic acid binding to be beyond the scope of this already extensive manuscript, we agree with the reviewer that the usage of sialic acid moieties is an important feature of several adenoviruses. We therefore provide a new alignment (supplemental figure 6) to demonstrate where key amino acids involved in sialic acid binding are (and are not) conserved, and comment on this in the revised discussion (lines 371-384):*

“The final previously described adenovirus fiber-knob receptor, which we have not addressed extensively in this study are sialic acids, as part of glycosylation motifs. Several Adenoviruses have been shown to bind to sialylated glycans, including HAdV-D37^{33,39}, HAdV-G52^{31,62}, and canine adenovirus serotype 2 (CAV-2)⁵⁹. Each of these three viruses binds to sialic acid by different mechanisms (Suppl. 6). Supplemental figure 6 shows that HAdV-D26/48K do not share the sialic acid binding residues found in HAdV-G52K or CAV-2 but do share the Tyr-142 and Lys-178 contact residues with HAdV-D37K. Further, the HAdV-D37K contact residue Pro-147 is between the sialic acid and the main chain oxygen which is functionally identical at the similar position in HAdV-D26/48K. Taken together, it remains plausible that HAdV-D26/48K may be capable of binding sialic acid in an HAdV-D37K like manner. However, binding does not equate to functional infection, as seen with HAdV-D19p³³ and further studies are required to ascertain whether HAdV-D26/48 are capable of utilising sialic acid to generate a productive infection. Further, HAdV-D37 was shown to require a specific glycosylation motif (GD1a) in order form a functional infection, so any assessment of sialic acid as an adenoviral receptor must be in the context of its glycan carrier³⁹.”

3) The computational calculations of binding energies between the fiber knob and receptor molecules is not convincing. For instance, it appears that there are a number of steric clashes in the complex between the CAR and Ad26K (green) and Ad48K (cyan), as the loops running into the CAR surface (Fig. 6B). If there are indeed such steric clashes, then the bar graphs shown in the Figures 5D and 8A are not meaningful. Additionally, based on the Fig. 5D, there appear to be stronger or comparable interactions between CAR and Ad26K/Ad48K in comparison to Ad5K (Fig. 5D), which is not convincing and supports my point of view that there could be some steric clashes.

Author response: *The reviewer is correct in her/his interpretation of this figure, which suggests an interaction is possible specifically at the α -interface, where no clashes exist. The clashes which sterically inhibit interaction between the fiber knob domain of Ad48/26 occur at the β -interface, which sterically limits the interaction with CAR. Because of these clashes, we predict the viruses are*

likely to have lower CAR binding affinities. To clarify this point, we have added the following text (see lines 194-204).

“Binding energies were calculated between the modelled fiber-knob proteins and CAR, restricting the calculation to only the α -interface to best model the conserved region. For the modelled complexes, a stable α -interface was predicted for all complexes modelled, albeit weaker for the known non-CAR utilising HAdV-B35K (Fig.5D) which has lower sequence conservation with HAdV-C5K. However, the interaction is complicated by a second CAR interface, termed the β -interface (Fig. 6A). The loops forming the β -interface with CAR-D1 differ between HAdV-C5, HAdV-D26, and HAdV-D48 fiber-knob (Fig. 6B). The shorter HAdV-C5K DG loop does not clash with the CAR-D1 surface, whereas the extended HAdV-D26K DG loop forms a partial steric clash, with surface seen to clash with the aligned CAR-D1, and HAdV-D48K DG loop is seen to form an even larger steric clash. Whilst the longer loop of HAdV-D26K is expected to be more flexible than that of HAdV-C5K, the HAdV-D48K DG loop is surprisingly stable due to the characteristics described (Fig.4C).”

In addition to the above text, we have performed additional, definitive surface plasmon resonance studies to directly quantify binding affinities for each of the described fiber knob proteins with CAR, CD46 and DSG-2, which appear to support our observations made through these modelling studies.

Reviewer 2

The manuscript by Baker et al. presents two new human adenovirus type fibre head structures, both from the D family. Both these adenoviruses are therapeutically important. The structural work appears to be performed well, and the structures are analysed in detail. The structures suggest low or no binding to the common human adenovirus receptors CAR and CD46, which is confirmed experimentally. In the case of HAdV-D26, these results strongly contradict those in reference 17. Ideally, this should be further investigated, but the information provided here is already important for the field, even if the real receptors have not been identified.

Author response: *we thank the reviewer for their positive comments and stating that “the structural work appears to be performed well, and the structures are analysed in detail”. We agree that our findings are somewhat controversial and strongly contradict some previous literature in the area (especially reference 17). As the reviewer points out, these are therapeutically important agents, and therefore warrant a detailed analysis of the basic virological mechanisms underpinning cellular infection, as we have attempted to execute here. To address the reviewers point that “ideally, this should be investigated further” (in relation to the interaction with CD46, our observations around (lack of) CD46 interactions mediated by Ad26/48 are now strengthened in the revised manuscript by the inclusion of additional surface plasmon resonance studies, which demonstrate a lack of direct interaction between hAdV-D26K or hAdV-D48K with CD46 (see figure 8E), thus validating our modelling methodology.*

Some human and non-human adenoviruses have been shown to bind carbohydrate receptors, and this could perhaps also be investigated, to start off, by a glycan array, for example. In my opinion, the manuscript is already very interesting to the adenovirus community and has implications for a wider therapeutical field.

Author response: *We appreciate the reviewer’s comment, and again thank the reviewer for their positive comment that “the manuscript is already very interesting to the adenovirus community and has implications for a wider therapeutical field”. The comment regarding potential interactions with cellular glycans and sialic acids is addressed in our response to reviewer 1. We have not performed*

an extensive analysis for these interactions, but provide a new alignment (supplemental figure 6) to demonstrate where key amino acids involved in sialic acid binding are (and are not) conserved, and comment on this in the revised discussion (lines 371-384).

As ever, some additional experiments could make it even more interesting, but I feel it is up to the Editors to decide if these would be necessary, because the current version is already internally consistent (i.e. no unsupported claims are made).

Author response: *We thank the reviewer for this positive comment.*

I also have the following, minor, suggestions to improve the manuscript:

- Indicate more clearly that the paper is about human adenoviruses.

Author response: *We have attempted to clarify this throughout, and using the revised nomenclature as suggested below, it should now be clearer that the entire manuscript relates to human adenoviruses – except for two mentions of canine adenovirus serotype 2 in the revised version (lines 376 and 378).*

- In a general journal like Nature Communications, a brief introduction to overall adenovirus structure and the fibre structure with a figure (spliced into Fig 1 if necessary), would be welcome.

Author response: *We have included a schematic of adenovirus as part of the revised figure 1, and now include the following brief introduction on adenovirus structure on lines 73-75*

“The fiber-knob is the receptor interacting domain of the fiber protein, one of three major capsid proteins along with the Hexon and Penton, as shown schematically in figure 1A.”

- I personally don't like abbreviating adenovirus to Ad. Could be written in full each time, it is not a long word. HAdV, HAdV-C5 etc. is also better, according to current taxonomic standards. The same for NABs, perhaps better written in full each time. But the Editors could say what they prefer.

Author response: *We thank the reviewer for highlighting this, and as per the reviewer's request, we have modified this to reflect the taxonomic standards throughout. The only exception where we were unable to achieve this was extending the dendrograms presented in figure 1 to reflect the full nomenclature, which resulted in an illegible figure. We therefore used the shortened nomenclature and added the following text to the figure legend... “for readability simple nomenclature is used, all are human adenovirus”...We hope this is acceptable for the reviewer.*

- The supplemental crystallographic table has a missing parenthesis and lines with different fonts and sizes.

Author response: *We apologise for this oversight and have amended this in the revised version.*

- When discussing the stability of loops (e.g. around line 136), possible stabilisation by crystal packing interactions should be analysed and discussed.

Author response: *We have included the following statement to cover this point (see revised manuscript, lines 163-7), and the calculated non-biological interaction energies are reported in supplemental figure 4, importantly crystal interactions do not impact our assessment for the biological interactions:*

“Crystal contacts did not reveal any specific interactions between these DG-loops and neighbouring non-trimer copies. We calculated the energy of interaction to be below the background threshold ($> -3.0\text{Kcalmol}^{-1}$) for all loops except DG. The DG-loop of HAdV-D26 is calculated to have an interaction

energy of -6.5Kcalmol^{-1} in two separate stretches of this exceptionally long loop (Suppl. 4). Importantly, no strong contacts are found within the inter-monomer cleft.”

52: increasedD

63: delete comma?

220: interactionS

277-278: needs citations (i.e. repeat those given earlier)

410: PyMoll should be PyMol

416-418: can be written as a sentence.

Author response: All of the above minor corrections have been addressed, and we thank the reviewer for their diligence in noting these.

Reviewer 3

The work presented by Baker et al. describes the structure of the HAd26 and HAd48 fibre knobs and compares them from a phylogenetic and structural point of view with fibre knobs of other adenovirus serotypes. These two adenovirus serotypes represent a major interest in therapy due to their developments in clinical trials in the vaccination field against Ebola or HIV. Modelling approaches for interaction with two out of the three main protein receptors CAR and CD46 (the third one DSG2 being simply mentioned since the structure of the complex is not yet available). Overall, the work is well presented and brings important new elements concerning the interaction of HAd26K and HAd48K with CAR. The length and rigidity of the DG loop of the fibre would be responsible for an interaction of 15 to 500 times less important than for the HAd5 used as a reference in this study. Rare points of contact would be possible with CD46, suggesting that other receptors may be needed for these two serotypes, which is important for understanding the tropism of these vectors of therapeutic interest.

Author response: We thank the reviewer for their positive comments regarding our manuscript, for describing it as being “well presented” and for noting that this is “important for understanding the tropism of these vectors of therapeutic interest.”.

- Although the competition data is convincing, it would have been interesting to have direct interaction data between the heads and CAR and CD46.

Author response: We agree with the reviewer, and have performed additional studies as suggested to generate direct interaction data, in the form of surface plasmon resonance studies. These findings are now presented as Figure 6D,E (for interaction between recombinant knob proteins and CAR), Figure 8D, E (for interaction between recombinant knob protein and CD46) and figure 9 (for interaction between recombinant knob protein and DSG2). Additional highlighted sections have also been added to the discussion section to expand upon the relevance of these important studies. We thank the reviewer for this excellent suggestion which we feel has helped to strengthen the manuscript significantly.

- A discussion stating if DSG2 is a rejected hypothesis for HAd26 and HAd48 would mean that new receptors remain to be identified. This point would be reinforced in the discussion section.

Author response: Again, we agree with the reviewer, and thank the reviewer for highlighting this important point. We are unable at the present time to model any potential interaction between HAdV-D26K and hAdV-D48K and desmoglein 2 (DSG2), since no high resolution complexed structure

is presently available for this interaction. The best known hAdV known to bind DSG2 is hAdV-B3, but the structure of the complex remain to be resolved. To address the reviewer's point as best possible, we include SPR analysis to determine potential interactions between hAdV-D26K/hAdV-D48K and DSG2 (and a control fiber knob protein from hAdV-B3) and include this data as the revised Figure 9. Our findings demonstrate that is no binding between hAdV-D26K/hAdV-D48K and DSG2, and therefore hAdV-D26 and hAdV-D48 are unlikely to use DSG2 as a cellular receptor. We agree that this indicates that new receptors are likely responsible for cellular uptake and infection, and reiterate this in the revised discussion (lines 396-8)...

“Together, our findings indicate that both hAdV-D26 or hAdV-D48 likely engage undocumented primary receptors to mediate cellular attachment and infection that remain to be elucidated”

These findings have again strengthened our manuscript and we are grateful to the reviewer for their helpful suggestion.

Specific editorial comments:

In particular, a revised manuscript would need to provide additional evidence for the suggested Adenovirus-receptor interactions, along the lines suggested by reviewer 1 (particular point 3) and reviewer 3 (particular point 1).

Author response: *We thank the editor for their thoughtful steer on our revisions, and we hope that the responses we have provided above are satisfactory to address editorial and reviewer concerns.*

Please make sure that these and all other concerns are addressed in full in a revised manuscript.

Author response: *We have provided an extensive and robust response to all concerns, and have significantly amended our manuscript. The resultant manuscript has been significantly improved for this process and we are grateful to the editor and the reviewers for their collective thoughtful comments.*

Reviewers' Comments:

Reviewer #2:

Remarks to the Author:

The authors have addressed all points raised and improved the manuscript. They have also performed some additional experimentation, although the definitive receptor preference of these adenoviruses has not been determined. Nevertheless I think this manuscript has significant interest.

Reviewer #3:

Remarks to the Author:

All the minor concerns raised in my previous review have been properly adressed.